# Selective Targeting and Enhanced Photodynamic Inactivation of Methicillin-Resistant *Staphylococcus aureus* (MRSA) by a Decacationic Vancomycin–Mesochlorin Conjugate

**DOI:** 10.3390/antibiotics14100978

**Published:** 2025-09-28

**Authors:** He Yin, Xiaojing Liu, Min Wang, Ying Wang, Tianhong Dai, Long Y. Chiang

**Affiliations:** 1Department of Chemistry, University of Massachusetts Lowell, Lowell, MA 01854, USA; heyin338@gmail.com (H.Y.);; 2Wellman Center for Photomedicine, Massachusetts General Hospital, Harvard Medical School, Boston, MA 02114, USA; 3Institute of Photomedicine, Shanghai Skin Disease Hospital, School of Medicine, Tongji University, Shanghai 200443, China

**Keywords:** combinatorial aPDT−antibiotics conjugates, antibacterial photodynamic therapeutics, decacationic photosensitizing agent, MRSA cell targeting, functionalized *meso*-chlorin

## Abstract

**Background/Objectives:** Covalent conjugation of an antibiotic vancomycin (VCM) moiety and a photosensitizing mesochlorin (*m*Chl_Pd_) unit into one molecular entity may present the potential to produce the combinatorial effect of both antibacterial photodynamic therapeutic (aPDT) and antibiotic activities. Our recent study indicated that a short linkage of <4 (C−C/or C−N) bond distances between these two moieties resulted in significant steric hindrance due to the bulky VCM, which greatly reduces the accessibility of the agent to the cell surface of methicillin-resistant *Staphylococcus aureus* (MRSA). The observed aPDT efficacy was found to be minimal. Here, we report that the revision of this linkage, via an EG_10_ unit using identical synthetic procedures, was able to resolve the issue. **Methods:** Accordingly, the corresponding combinatorial aPDT−antibiotic compound, consisting of two covalently bonded quaternary ammonium pentacationic arms on the mesochlorin chromophore core, designated as VCMe-*m*Chl_Pd_-N_10_^+^ (LC40e^+^), was prepared for applications in antibacterial photodynamic inactivation (aPDI) activity. It was selected to investigate its enhanced binding and targeting ability to the surface of Gram-positive MRSA cells. Subsequent antibacterial photodynamic therapeutic (aPDT) activity to inactivate MRSA was investigated to substantiate the corresponding cell-surface binding effect on the efficacy of aPDT. **Results:** We found that the covalent combination of 10 positive charges and an MRSA-targeting vancomycin (VCM) moiety in a conjugated structure, functioning as an antibiotic–decacationic photosensitizing agent (Abx-dcPS), was capable of largely improving the MRSA cell-targeting efficiency. Importantly, variation in the chain length of the oligo(ethylene glycol) linker of VCMe-*m*Chl_Pd_-N_10_^+^, which was sufficiently long enough to properly separate the photoactive mesochlorin ring moiety from the VCM moiety within the molecular structure, resulted in significantly enhanced aPDT activity. The new conjugate provided nearly complete eradication (>6.5-log_10_ colony-forming units (CFU) reduction) of MRSA cells in vitro. The aPDT efficacy followed the order Abx-dcPS (combinatorial decacationic) > dcPS (decacationic) >> nPS (nonionic). This order was also verified by the relative physical binding trend of these PSs using either nPS-, dcPS-, or Abx-dcPS-pretreated and pre-fixed MRSA cells in investigations of fluorescent confocal microscopy, UV–vis fluorescence spectroscopy, and transmission electron microscopy (TEM). **Conclusions:** Furthermore, the molecular conjugate of Abx-dcPS may provide covalent co-delivery of two drug components concurrently, which might also serve as an effective antibiotic agent after aPDT and potentially prevent the reoccurrence of MRSA-induced infection.

## 1. Introduction

Methicillin-resistant *Staphylococcus aureus* (MRSA) strains are capable of producing various toxins and enzyme proteins, which contribute to their high pathogenicity and make them potentially lethal to humans [1,2,3,4]. The persistent presence of these bacterial surface-binding factors on host cells can lead to a range of severe infections, particularly skin and soft tissue infections, as well as others. The continued emergence of MRSA variants, which adapt to different community settings and exhibit varying levels of resistance to conventional antibiotics, presents significant therapeutic challenges [5,6,7]. Although new antibacterial drugs are continually being developed, multidrug-resistant strains continue to arise, highlighting the urgent need to explore alternative therapeutic strategies [8,9,10].

Antimicrobial photodynamic inactivation (aPDI) is a chemical technique that uses photoresponsive organic chromophore compounds with low toxicity to activate a lethal photosensitization process that eventually causes bacterial cell death. In the light of continuing concern about the foreseeable upsurge of multidrug resistance in microbes upon treatment with conventional antibiotics, the organic photonic-based aPDI approach has secured increasing attention and appreciable recognition as an alternative treatment method for photokilling pathogenic microbial cells [11,12,13,14]. Importantly, this approach does not cause resistance activity [15,16]. In this technique, microbial cells are pre-impregnated with a photosensitizer (PS) molecule, consisting of effective transient excited triplet states, and subsequently exposed to a specific harmless visible light source with an emission spectrum bandwidth matching the major optical absorption wavelengths of PS. The applied photoenergy then sensitizes the PS, which leads to the generation of triplet energy from the corresponding photoexcited triplet ^3^PS* intermediate. Subsequent transfer of this triplet energy to triplet molecular oxygen (^3^O_2_) transforms it into singlet oxygen (^1^O_2_) as one of the reactive oxygen species (ROS) in the Type-II photomechanism, which exhibits cytotoxic effects on microbial cells. The products of the ROS generated in these reactions may induce various kinds of irreversible damage to components of microbial cells via oxidation of biomolecules, which alters the corresponding metabolic activities [17,18]. Accordingly, ROS-mediated toxicity can be independent of and non-responsive to alterations in antibiotic binding affinity caused by mutations in cell surface structures. In fact, aPDI-active PS neither induces microbial resistance nor is subject to the influence of existing drug resistance status. This explains the high therapeutic potential of aPDI against multidrug-resistant (MDR) microbes, such as MRSA.

Several classes of PS compounds have been developed and investigated over the years as antimicrobial inactivators [19,20,21,22,23,24]. Among them, porphyrin, benzoporphyrin, chlorin, and texaphyrin analogous derivatives are the most common PSs used in cancer and tumor therapy studies. In the case of aPDI, an additional class of polycationic fullerene (C_60_ or C_70_) monadduct or bisadduct derivatives has been demonstrated as photoresponsive PSs (C_60_/_70_PSs) in related treatments, with the potential application of O_2_-independent aPDI using an activated intermolecular or intramolecular electron (e^−^)-transfer photomechanism [25,26,27,28,29]. Moreover, considering the charge factor parameter and the upper limit of charge number as a threshold for achieving optimal efficiency in biocidal action—which may lead to loss of cell viability—certain explorations were carried out by us using photosensitizing cationic chlorin derivatives [30]. In this study, we applied a similar, well-defined decacationic strategy to target MRSA involving a combination of interactive H-bonding and surface static charge interactions as a multiple binding-sites approach to extend the binding area. This approach was proposed to partially overcome the chemical alternation and transformation of –*L*-Lys*-D*-Ala-*D*-Ala to –*L*-Lys*-D*-Ala-*D*-lactate by bacteria during mutation processes [31]. The targeting ability is based on the highly negatively charged nature of lipopolysaccharides (phosphate and carboxylic acid anions) at the outer membrane surface of the Gram-negative bacterial cell wall and anions of teichoic acid and terminal carboxylic acids of the outer thick peptidoglycan layer of the Gram-positive bacterial cell wall. It is in addition to the simulation of H-binding moieties of vancomycin to the bacterial cell wall [32,33]. Vancomycin was selected for this study for several reasons. Its chemical structure allows selective modification without compromising molecular integrity, due to the presence of a single primary amine that facilitates conjugation. It also exhibits strong targeting ability against methicillin-resistant bacteria such as MRSA, making it a reliable choice for our investigation. Furthermore, in addition to enhancing antibacterial photodynamic inactivation (aPDI), vancomycin may provide a delayed antibiotic effect after light treatment, offering the potential to prevent the recurrence of infections. The targeting combination by design and synthesis may significantly increase the number of multi-binding sites to most Gram-positive bacterial cells. Accordingly, we demonstrated the cell-targeting ability differentiation between two aPDI agent molecules, the decacationic photosensitizer (dcPS) Chl_pd_-N_10_^+^ (LC38^+^) and the antibiotic decacationic photosensitizer (Abx-dcPS) VCMe-*m*Chl_pd_-N_10_^+^ (LC40e^+^), as shown in Figure 1, on MRSA cells, using a neutral photosensitizer (nPS) Chl_pd_-EG_n_ (LC37) only serving as the reference compound for comparison of targeting ability.

## 2. Results

### 2.1. Material Preparation

Naturally occurring π-conjugated dye molecules, such as pheophytin, exhibit long-lived triplet excited states and can serve as photoactive cores and precursors for synthesizing new PSs for aPDI [11,34,35]. Among a number of known chlorophyll-related precursor substructures, we selected pheophytin (Phe), a natural dye, containing a photoactive pheophorbide (Pheo) core, as the starting material. It was generated by demetallation of chlorophyll-a, which is the main soluble organic dye component of chloroplasts in spinach leaves extractable by acetone. Its Pheo core structure consists of a conjugation of four pyrroles covalently linked by four methine bridges as a tetrapyrrole unit. The ring formation comprises a system of 18 π-conjugated electrons in the chromophore structure. It is highly photoactive and responsible for the absorption of light to provide energy for plant photosynthesis and the electron-transfer pathway.

We recently developed a key well-defined water-soluble pentacationic *N,N’,N,N,N,N*-hexapropyl-hexa(aminoethyl)amine-penta(quaternary methyl-ammonium iodide) as a functional sidearm, H_2_N(C_2_N^+^C_1_C_3_)_5_, along with its neutral precursor H_2_N(C_2_NC_3_)_5_ analogous [27]. The former contains a fixed number of five positive charges in a linear alkylamino chain. The latter unit consists of five in-chain tertiary propylethylamino groups and one primary amine end-group, suitable for carrying out the versatile amidation reaction with either a carboxylic acid, ester, or lactone functional group. Therefore, the unit of H_2_N(C_2_NC_3_)_5_ can be applied as a base for the incorporation of five cationic charges in a progressive increment. In the case of pheophytin a, there are two regions at the opposite side of the residual olefin on the central core appropriate for making structural modification. Interestingly, we found that the precursor arm H_2_N(C_2_NC_3_)_5_ reacts more readily at the phytyl (isoprenoid) ester moiety than the cyclic ketone ester moiety. The reaction is also insensitive to olefinic ring carbons that preserve the central Pheo core ring system. This allowed us to carry out a sequential amidation to substitute the alkyl phytol tail first prior to the more covalent H_2_N(C_2_NC_3_)_5_ arms attachment as sidechains onto a Pheo core in a regio-selective and kinetic control manner to add either one or two pentacationic arms. This versatile synthetic method enables functionalization of the porphyrin ring with a tunable number of cationic charges for evaluation of the structural relationship of charge factors to the targeting ability of resulting mcPSs toward infectious bacteria and aPDI efficiency.

The antibiotic vancomycin hydrochloride (VCM) molecule was reported to be capable of functioning as a cell wall synthesis blocker by binding to the C-terminal tripeptide (*L*-Lys-*D*-Ala-*D*-Ala) of the peptidoglycan precursor, especially on Gram-positive bacteria [36,37]. It is rational to apply VCM as an additional component to multications for the enhancement of bacterial targeting. The development of LC40e^+^ is based on an evolutionary series of mesochlorin conjugates. Our earlier research demonstrated that varying the number of positive charges (from 2 to 15) on the mesochlorin scaffold significantly impacts antibacterial activity, with 10 charges yielding the highest efficacy [30]. Building on these findings, vancomycin (VCM) was introduced as a synergistic targeting component to enhance aPDI. However, our recent study revealed that when the linkage between the mesochlorin core and VCM was shorter than four C–C or C–N bonds, the bulky VCM moiety caused substantial steric hindrance, greatly reducing MRSA surface accessibility and minimizing aPDT efficacy. To overcome this limitation, we revised the design by introducing an EG_10_ linker using identical synthetic procedures. This modification successfully alleviated steric hindrance. Consequently, we prepared LC40e^+^, a combinatorial aPDT–antibiotic compound consisting of two covalently bonded quaternary ammonium pentacationic arms on the mesochlorin chromophore core and a short polymer linker conjugated to VCM, for evaluation of antibacterial photodynamic inactivation (aPDI) activity. Successful linking of mChl and VCM components together in one covalent molecular conjugate at a distance from each other should add desirable independent molecular features that should also incorporate antibiotic activity to the aPDI photomechanism. In addition, a combination of multiple (ten) cationic charges and Gram-positive bacteria-targeting VCM in an integrated therapeutic agent was proposed to enhance the potential lethal inactivation of MRSA by LC40e^+^.

### 2.2. Photophysical Properties of dcPS Chl_Pd_-N_10_^+^ (LC38^+^) and Abx-dcPS VCMe-mChl_Pd_-N_10_^+^ (LC40e^+^)

aPDI activity was correlated with the photophysical properties of the photosensitizers, especially the value of the absorption extinction coefficient (ε) and the yield of subsequent fluorescence emission at the aPDI practicing wavelength. In terms of chlorin derivatives, their aPDI efficacy was well-studied and recognized as a function of photoinduced singlet oxygen (^1^O_2_) generation via triplet state in the first step of biological cascade sequences leading to bactericidal effects [11,12,13,14,17,18]. In general, steady-state optical absorption of chlorin derivatives was based on an 18 π-conjugated ring system of either a chlorin (LC37 and LC38^+^) or mesochlorin (LC40e^+^) core. In the case of decacationic chlorin (LC38^+^), a close resemblance of peak profiles as those of oligoglycolated chlorin (LC37) was obtained, showing the main Soret (as S_o_ → S_2_ transition) absorption band centered at 408 nm along with two *Q*_*x*_ transition bands at 505 and 536 nm and two *Q*_*y*_ transition bands (as S_o_ → S_1_ transitions) at 609 and 666 nm in Figure 2A(a,b). Among them, the extinction coefficient value of the latter 666 nm band was measured as 39% in intensity as that of the former 408 nm band. All these photoinduced π–π* transitions can be correlated to the energy gap between the highest occupied molecular orbitals (HOMO) and the lowest unoccupied molecular orbital (LUMO) orbitals of dcPS and Abx-dcPS. In the case of LC40e^+^ (Figure 2A(c)), the Soret and *Q*_*x*_ bands were broadened significantly as compared with those of LC37 and LC38^+^, but with higher extinction coefficients in the *Q*_*x*_ band region when the profile was normalized at the 408 nm band. The most visible change occurred at the *Q*_y_ band region, where the ε value of the 665 nm band was found to be reduced by roughly 51% in intensity as compared with that of LC38^+^. It is likely owing to the loss of one conjugative olefin bond at the C_3a_–C_3b_ position of the mesochlorin ring moiety (Figure 1), giving a slightly less overall conjugation length. Nevertheless, this peak intensity is higher than that of VCM-Chl_Zn_-N_5_^+^ (LC39) since the VCM moiety was extended further away from the chlorin ring by an EG_n_ linker to prevent partial disruption of long-range π–π* transitions and co-planarity of the mesochlorin ring system [30].

### 2.3. MRSA Cell Targeting Evaluation of dcPS Chl_Pd_-N_10_^+^ (LC38^+^) and Abx-dcPS VCMe-mChl_Pd_-N_10_^+^ (LC40e^+^)

Appropriate fluorescence emission of chlorin and mesochlorin moieties will allow us to detect their presence on biological cells. Due to the fact of strong blue light absorption at 405–410 nm, we selected this wavelength range for the study of photoexcitation evaluation in either spectroscopic measurements or biological experiments. Under blue light irradiation, two fluorescent emission peaks centered at 467 (blue emission) and 676 nm (red emission) were observed, with the latter (Figure 2B(a–c)) being much more intensive in extinction coefficient (ε). Accordingly, the red fluorescent emission was applied as the primary imaging detection in the confocal microscopic measurements, while the blue fluorescent emission served as the secondary supplement. We first evaluated the ability of either dcPS or Abx-dcPS to bind to either the MRSA IQ0064 strain or host HaCaT cells, with the latter as the model of comparison study. Experimentally, both MRSA and HaCaT cells, with or without the pretreatment of either dcPS or Abx-dcPS, were prefixed with glutaraldehyde and suspended in acetone. During this process, three 2-amine moieties of LC40e^+^ may also react with glutaraldehyde to give the corresponding enamines. However, the closest one of these enamines is located at a distance of three C–C single bonds away from the chlorin ring region. Therefore, it should have no effect on the FL characteristics of LC40e^+^. In the case of LC37 and LC38^+^, no reactive center exists in their molecular structure. A small portion of cell suspension was directly deposited on a supporting carbon−copper film grid (200 mesh), followed by solvent removal via evaporation, and dried in vacuo to afford the sample plate for transmission electron microscopic (TEM) micrographic investigation of morphological changes in the bacterial cell.

TEM images of untreated MRSA clusters revealed smooth cell membranes (Figure 3(a1,a2)) without visible peptidoglycan residues or attached biomolecules, resulting in a uniform morphology. This made it straightforward to allow differentiation of the membrane surface binding among LC37 (nPS, Figure 3(b1,b2)), LC38^+^ (dcPS, Figure 3(c1–c3)), and LC40e^+^ (Abx-dcPS, Figure 3(d1–d3)) by the morphology comparison. As a result, a minimum binding of LC37 was observed with only a small number of bacteria clusters showing the attachment of molecules (Figure 3(b2)). This is consistent with the fact that LC37 comprises two neutral oligo(ethylene glycol) chains and only two cationic amino end-moieties that give weak MRSA affinity. As the number of cationic quaternary methyl ammonium iodide moieties increases to ten per molecule (decacationic) in LC38^+^, a progressive increase in the agent attachments was detected in Figure 3(c2,c3), revealing an enhanced ability to be attracted at the cell membrane surface. With an additional VCM moiety other than decacationic charges in the structure of LC40e^+^, the most membrane surface regions of bacteria clusters were covered by Abx-dcPS, as shown in micrographs of Figure 3(d2,d3), to afford a slightly rough surface and a thin layer coverage of organic substance in less morphological contrast than the MRSA cell.

We also prepared ultra-thin section samples of MRSA cells and its nPS-, dcPS-, or Abx-dcPS-pretreated cells using bladed microtome slices, where the cells were embedded in a matrix of Epon t812 polymer. They were collected on uncoated copper grids (200 mesh) and stained with uranyl acetate and lead citrate prior to the TEM measurements. By comparing with the intercellular membrane morphology between MRSA cells (Figure 3e) and LC40e^+^-pretreated MRSA cells (Figure 3f), it was unambiguous to detect many LC40e^+^ conjugates at different locations, either surrounding the close vicinity or directly attached to the MRSA cell membrane, as shown in Figure 3f. Nevertheless, it was not possible to differentiate the existence of LC40e^+^ in the intracellular space due to the lack of clear image contrast. A combination of TEM with other techniques such as flow cytometry, fluorescence or confocal Z-stack imaging of biofilms, extracellular polymeric substance (EPS) disruption assays, atomic force microscopy, and mass spectrometry may allow us to observe intracellular localization of LC40e^+^. It is to be applied in the future investigation. In the present study, TEM has provided high-resolution images of the cell surface, allowing us to have a more intuitive visualization of drug distribution at the cell boundary. Conclusively, these TEM micrographic results physically substantiated our hypothesis that a combination of decacationic charges and an MRSA-targeting VCM moiety to chlorin as a covalently conjugated derivative may significantly increase its ability to be attracted to the membrane surface of MRSA cells, giving the affinity order of Abx-dcPS > dcPS >> nPS.

Chemically, the relative attachment or binding quantity among nPS, dcPS, and Abx-dcPS on MRSA cells was further validated by UV-vis and fluorescence spectrum measurements of agent-pretreated bacterial cell samples. As shown in Figure 2C, the optical absorption peak intensity of all Soret, *Q*_*x*_, and *Q*_y_ bands at 405–415, 505–540, and 660–670 nm, respectively, displayed a progressive increase going from none for the MRSA cells alone sample (Figure 2C(a)) to low for the LC37-pretreated MRSA cell sample (Figure 2C(b)) and moderate for the LC38^+^- (Figure 2C(c)) and LC40e^+^-pretreated MRSA cell samples (Figure 2C(d)). They were indicative of either dcPS or Abx-dcPS molecular association with MRSA cells.

These optical absorption characteristics were accompanied by corresponding fluorescent emissions at 658–680 nm (Figure 2D), upon photoexcitation at 410 nm. Quantitatively, the cell count per mL in each solution of Figure 2D was carried out by the flow cytometric analysis technique. The measured extinction coefficient (ε) value of each mL solution with MRSA cells was correlated to that of a pure agent (without MRSA) in a known concentration to calculate the effective number of agent molecules per mL present with MRSA cells. They were then divided by the cell count per mL to obtain the normalized ε value of fluorescent emission and the relative molar quantity of either nPS, dcPS, or Abx-dcPS present on MRSA cells. As a result, Figure 2D indicated a clear significant enhancement in peak intensity of chlorin-based fluorescent emission going from moderate (100% as the reference) for the LC37-pretreated MRSA cell sample (Figure 2D(b)) to 155% in relative intensity for LC38^+^- (Figure 2D(c)) and nearly 220% in relative intensity for the LC40e^+^-pretreated MRSA cell sample (Figure 2D(d)). These were corresponding to the attachment of LC38^+^ and LC40e^+^ on MRSA cells being 1.55- and 2.2-fold in molar quantity on average to that of LC37. Since all agent-pretreated MRSA cell samples were centrifuged and washed repeatedly to remove an excessive amount of free agents in solution during the preparation procedure, all observed fluorescent emission should be associated with the chlorin-based aPDI agents being bound on the cell membrane surface and accountable for their relative quantities. Accordingly, Figure 2D should provide direct evidence of LC38^+^ and LC40e^+^ molecules being targeted on MRSA cells.

We further studied the binding characteristics of LC37, LC38^+^, and LC40e^+^ using confocal fluorescence microscopy (CM). Under CM condition of 405 nm irradiation (LED light, 30 mW), all present nPS, dcPS, and Abx-dcPS exhibited two similar fluorescent emission bands centered at 490 (minor, blue) and 670 nm (major, red). Therefore, the detector was set with the emission shutter in a wider wavelength range to capture these two emissions. Since the red emission can only be possible to occur from chlorin- and mesochlorin-based agents, it should be more accurate to assess the relative targeting quantity on MRSA cells. As a result, no red emission was detectable on the sample plate with MRSA cells alone (Figure 4(a2)), as expected. The red emission became slightly visible on the sample plate with LC37-pretreated MRSA cells (Figure 4(b2)) and turned clearly detectable with LC38^+^-pretreated MRSA cell samples (Figure 4(c2)). It was significantly enhanced on LC40e-pretreated MRSA cell samples (Figure 4(d2)), showing the relative cell affinity order of Abx-dcPS > dcPS >> nPS. It is consistent with the observed ε value (Figure 2D) of LC38^+^ and LC40e^+^ on MRSA being 1.55- and 2.2-fold higher than that of LC37. In the case of blue emission, MRSA cells also showed such ability (Figure 4(a3)); in overlapping with those of PSs, it became the reference for supplemental identification of MRSA cell cluster morphology. The overlay of the first three columns of Figure 4, including bright field, red fluorescent emission (640–780 nm), and blue fluorescent emission (450–550 nm) images, provided clear accounts to conclude the successfully enhanced attachment of Abx-dcPS (LC40e^+^) on MRSA cells by a combination application of decacationic charges and an MRSA-targeting VCM moiety. The observation was also consistent with those conclusions found by TEM micrographic and UV-vis spectroscopic data.

In the case of the host HaCaT cell samples in anhydrous acetone solution, with or without either dcPS or Abx-dcPS pretreatment, for the UV-vis spectroscopic measurements, the preparation method followed those used in the procedure of making dcPS-pretreated MRSA cell solutions, as described above. Interestingly, none of the optical absorption spectra of LC37- (Figure 5A(b)), LC38^+^- (Figure 5A(c)), and LC40e^+^-pretreated (Figure 5A(d)) host cells exhibited meaningful absorption bands arising from the chlorin-based moiety of these dcPS agents, the same as the featureless spectrum of host cells alone (Figure 5A(a)). These revealed either the absence or lack of either dcPS or Abx-dcPS molecules being associated with the host cells. Further, the corresponding FL spectra, as shown in Figure 5B, displayed only insignificant emission at 655–680 nm by judging the large noise level of all spectra and the intensity counts of only ~40, ~270, and ~150 for LC37- (Figure 5B(a)), LC38^+^- (Figure 5B(b)), and LC40e^+^-pretreated (Figure 5B(c)) host cell samples, respectively, that were significantly below the intensity count number shown in Figure 2D with MRSA cells. Therefore, it is appropriate to exclude the possible attachment of either dcPS or Abx-dcPS molecules to the host cell that revealed the rather low binding affinity.

In addition, we also prepared the host HaCaT cell samples by culturing them on glass slide plates that were followed by staining with the adenine–thymine binding 4′,6-diamidino-2-phenylindole (DAPI) as a blue fluorescent stain dye to the DNA nuclei of host cells. These slide plates were applied in the confocal microscopic imaging evaluation, with the results displayed in Figure 6. Under irradiation at 405 nm using an LED light (30 mW), no red emission at 640–780 nm was detected on LC37-pretreated host cell sample (Figure 6(b3)), indicating negligible binding of LC37 on the host cell. In the case of LC38^+^- and LC40e^+^-pretreated host cell samples, nearly none or only a few single spots of red emission were observed in Figure 6(c3,d3), respectively. Most of these red spots were not even associated directly with the cell membrane surface, perhaps as the aggregated impurity clusters precipitated during the centrifuge process. Since the red fluorescent emission is directly indicative of the existence of chlorin- or mesochlorin-based aPDI agents, the results of Figure 6 allowed us to conclude either none or negligible binding of nPS, dcPS, and Abx-dcPS molecules on the host cell, consistent with the same phenomena reached by the analysis of Figure 5B. Therefore, the observation allowed us to propose that the minimum photocytotoxicity to the host cells may be expected during the practice of aPDI using these dcPSs.

### 2.4. Reactive Oxygen Species (ROS) Generation Efficiency of dcPS Chl_Pd_-N_10_^+^ (LC38^+^) and Abx-dcPS VCMe-mChl_Pd_-N_10_^+^ (LC40e^+^)

A highly fluorescent tetrasodium α,α’-(anthracene-9,10-diyl)bis(methylenemalonic acid salt) (ABMA) was synthesized previously and used as a reactive probe in the solution detection of singlet oxygen (^1^O_2_) generated by either LC38^+^ or LC40e^+^ [28,38]. The neutral LC37 was included only as a reference for comparing targeting ability. It demonstrated negligible targeting toward MRSA. Since effective aPDI requires not only ROS generation but also sufficient photosensitizer localization at the target site, compounds with minimal targeting ability are unlikely to achieve meaningful inactivation, regardless of their ROS yield. Therefore, further evaluation of ROS generation by LC37 was not pursued, as its lack of targeting rendered it unsuitable for effective aPDI. Therefore, ROS evaluation was not conducted for this compound. Facile chemical trapping of ^1^O_2_ by ABMA molecules results in the formation of the corresponding non-fluorescent 9,10-endoperoxide product ABMA-O_2_. Accordingly, the effective amount of fluorescence emission loss in intensity at 428 nm, owing to the chemical translation in structure, can be correlated to the same molar quantity of ^1^O_2_ produced by either LC38^+^ or LC40e^+^. It is valid because of the high reaction kinetic rate of the in situ trapping process by the fluorescent probe in co-existence with the PS molecules in the same experimental solution that allowed us to assume the internal decay rate of ^1^O_2_ being minimum. Production of ^1^O_2_ originated from the intersystem crossing process from the singlet excited state of ^1^(Chl_Pd_-N_10_^+^)* and ^1^(VCMe-mChl_Pd_-N_10_^+^)* to their corresponding triplet excited state of ^3^(Chl_Pd_-N_10_^+^)* and ^3^(VCMe-mChl_Pd_-N_10_^+^)*, respectively, followed by the Type-II triplet energy transfer to O_2_.

Consequently, under all three light sources of white LED, 400 nm LED, and 650 nm laser used for the irradiation of dcPS (LC38^+^) and Abx-dcPS (LC40e^+^), rapid loss of the fluorescent emission intensity of ABMA was observed in Figure 7A,B, respectively. An ABMA probe solution without dcPS was evaluated for the comparison purpose. Apparently, slow slight photodegradation of the ABMA molecule in DMF–H_2_O (1:9, *v*/*v*) was observed (Figure 7A(d–f),B(d–f)) depending on the effective irradiant fluence of light applied on the solution, where white LED (2.0 W) light was higher than those of 400 nm LED (320 mW) and 650 nm laser (200 mW) lights. Similar fluorescent emission intensity decreasing trends were detected with the decreasing order by Figure 7A(a),B(a) > Figure 7A(b),B(b) > Figure 7A(c),B(c), based on the curve-slope change. The nonlinearity of all curves may be caused by a progressive decrease in the ABMA concentration in solution. By comparison between either Figure 7A(a),B(a) or Figure 7A(b),B(b), the fluorescence intensity decreasing rate was faster for the latter, reaching nearly zero within 75 or 125 sec of irradiation, respectively, indicating a more rapid and higher production quantity of ^1^O_2_ for LC40e^+^ than LC38^+^ under the same light wavelength and irradiance exposure. These photophysical characteristics should lead to better performance of LC40e^+^ in aPDI efficacy than LC38^+^.

We previously reported the possibility to achieve intramolecular photoinduced electron transfer from counter iodide anions to the photoexcited fullerene cage moiety of water-soluble C_60_[>M(C_3_N_6_^+^C_3_)_2_]-(I^−^)_10_ leading to, eventually, the formation of O_2_^−^**·** [28]. It was also previously reported that the iodide anion (I^−^) is capable of initiating a reaction with ^1^O_2_ to form several types of ROS, including O_2_^−^**·**, H_2_O_2_, and HO**·** [39]. The reaction was associated with a plausible mechanism of electron-transfer event from I^−^ to ^1^O_2_ that led to the possible formation of O_2_^−^**·**. Accordingly, we performed similar experiments to monitor the production of O_2_^−^**·** by the same series of mcPSs to address whether it is the instance. In the O_2_^−^**·**-trapping reaction, a regioisomeric mixture of superoxide radical-reactive fluorescent probe, potassium bis(2,4-dinitrobenzenesulfonyl)-2′,4′,5′,7′-tetrafluorofluorescein-10′(or 11′)-carboxylate isomers (DNBs-TFFC), was used for the experiment. High reaction selectivity of this probe toward O_2_^−^**·** was reported with a O_2_^−^**·**/^1^O_2_ sensitivity ratio of 46 [40]. However, all compounds we selected for evaluation induced roughly similar intensities of fluorescence from the probe, with fluctuation of only ±1% from that of the blank control. Therefore, we concluded that no significant production of O_2_^−^**·** was observed regardless of the variation in mcPSs.

### 2.5. aPDI Efficiency of Chl_Pd_-N_10_^+^ (LC38^+^) and VCMe-mChl_Pd_-N_10_^+^ (LC40e^+^) Against MRSA Cells

Photocytotoxicity efficacies of nPS (LC37), dcPS (LC38^+^), and Abx-dcPS (LC40e^+^) against MRSA cells in vitro were evaluated. As a control, vancomycin alone at the equivalent dose presented a negligible effect on the viability of MRSA cells, as shown in Figure 8A(a),B(a). This is owing to the fact that all aPDI experiments were carried out within a relatively short period of time (14−80 min), well below the kinetic rate required for eliciting antibiotic activity. Therefore, a VCM moiety was applied for enhancing its targeting ability to MRSA cells at the initial stage of aPDI, as we concluded during the targeting evaluations using the UV-vis-FL spectroscopic technique and fluorescent confocal microscopic imaging analyses stated above. We found that MRSA cells were susceptible to light-mediated killing by dcPS and Abx-dcPS having 10 cationic charges. The trend was obvious when the comparison of survival fraction log_10_ CFU values (*p* < 0.001) obtained for noncationic LC37 (Figure 8A(b),B(b)) and decacationic LC38^+^ (Figure 8A(c),B(c)) showed an increased photokilling rate for the latter under the same conditions of light exposure, irradiance, and drug concentration at both 405- and 660 nm photoexcitation. With the covalent attachment of a VCM moiety to the molecular structure of LC38^+^ at the C_3b_ position along with an EG_n_ linker (Figure 1) to result in the agent structure of LC40e^+^, a large enhancement of photokilling efficacy with higher log_10_ CFU reduction values (*p* < 0.001) was detected in Figure 8A(d) (λ_ex_ 660 nm) and Figure 8B(d) (λ_ex_ 405 nm), where complete eradication (>6.5-log_10_ CFU reduction, Figure 8A(d)) was possible using a concentration of 10–20 µM with a 115 J/cm^2^ exposure of 660 nm light. In the case of 405 nm irradiation, a 5.0-log_10_ CFU reduction was achieved at the same concentration range with a lower light exposure of 20 J/cm^2^ (Figure 8B(d)).

### 2.6. aPDI Efficiency of VCMe-mChl_Pd_-N_10_^+^ (LC40e^+^) Against MRSA Biofilms

Microbial biofilms are responsible for a variety of local topical infections. Characteristics of bacterial communities-based biofilms, attaching to each other and firmly to substratum surfaces by bacterial adherence, may exhibit different features and responses to the aPDI treatment. It is owing to the potential surface coverage of bacterial cell-produced extracellular polymeric matrix that increases the difficulty for aPDI agents to penetrate efficiently and reach the pathogenic cells entrenched underneath. This may also intensify the resistance to antibiotics and other antimicrobial treatments and means [41]. Accordingly, it was of interest to us to investigate the possible deviation of aPDI efficacy when treating 24 h-old biofilms under the similar aPDI conditions applied above. Based on the results of aPDI efficiency against planktonic MRSA shown in Figure 8, LC37 and LC38^+^ did not display promising activity due to the lack of selective targeting antennas, in contrast to LC40e^+^. Therefore, only LC40e^+^ was selected for evaluation of its aPDI efficiency on biofilms. With the light exposure of 115 J/cm^2^ at 660 nm irradiation, a roughly 2.0-log_10_ CFU reduction was observed using an LC40e^+^ concentration of 20 µM (Figure 9A(b)), whereas the dark control gave only a 0.5-log_10_ value (*p* < 0.05, Figure 9A(a)) at the same concentration. The former was at least 4.5-log_10_ less than that obtained in non-biofilm experiments carried out above. When a much less light exposure of 20 J/cm^2^ at 405 nm (Figure 9B(b)) was applied, only a 1.0-log_10_ CFU reduction (*p* < 0.05) was revealed in Figure 9B(b) using an LC40e^+^ concentration of 20 µM. Although LC40e^+^ achieved significant activity against biofilm-associated MRSA (~2-log reduction), its efficacy seemed to be lower compared with planktonic cells (>6.5-log reduction). It should be noted that in a dose-dependent experiment, higher inactivation of biofilms can be achieved by applying higher aPDI doses. This outcome is consistent with the inherent resistance of biofilms due to their dense extracellular polymeric substance (EPS) matrix and restricted photosensitizer penetration. Future studies will focus on improving biofilm eradication, including penetration analysis by CLSM z-stack imaging, EPS disruption assays to assess matrix destabilization, and combination approaches with biofilm-dispersal agents.

### 2.7. Cytotoxicity of LC40e^+^ Mediated aPDI to Human Cells

We also investigated the potential photocytotoxicity to human cells under the same aPDI experiment conditions as described above. Consequently, at the Abx-dcPS concentration of 5.0 µM, MRSA cell CFUs were reduced by over 5-log_10_ with the light exposure of 115 J/cm^2^ at 660 nm irradiation. In contrast, the viability of HaCaT cells was reduced by only about 0.4-log_10_ (*p* < 0.001) under the same experimental conditions, as shown in Figure 8A(e). The difference indicated clearly that Abx-dcPS exhibited higher selectivity in targeting MRSA than HaCaT cells, which is consistent with the much less cell membrane attachment ability of LC40e^+^ to HaCaT than MRSA cells, as concluded above by cell morphology analyses. Similar results were also observed under the light exposure of 20 J/cm^2^ at 405 nm irradiation, as shown in Figure 8B(e) (*p* < 0.001). However, at higher concentrations of Abx-dcPS (10–20 µM) in the former figure, certain photocytotoxicity of HaCaT cells was also detected, giving the corresponding viability reduction to about 1.0-log_10_ while the MRSA cells were nearly eradicated. It is worth mentioning that an effective aPDI process relies on three key factors: the light source, photosensitizers, and molecular oxygen. In this study, we initiated the evaluation at a relatively lower light dosage combined with a higher photosensitizer concentration to ensure the photosensitizers’ potential effectiveness. The concentration can be further reduced by increasing the light dosage or prolonging the irradiation time. Therefore, optimal concentration of aPDI agent and light exposure should be investigated in future study.

## 3. Discussion

MDR bacterial infections have been commonly recognized as one of the major causes of disease and death worldwide. The rise in antibiotic resistance poses a global health emergency. Accordingly, innovative non-antibiotic-related therapeutics serving as alternative treatment techniques for microorganism-caused infectious diseases are desirable for combating emerging or reemerging antibiotic-resistant pathogens. Since the photomechanism of PS involves non-specific attack by ROS on molecular components of the cell membrane, resistance is unlikely to develop. Therefore, the approach becomes our main focus of the study. When a PS is derivatized by covalent functionalization to enhance its targeting ability to MDR bacteria, the photokilling efficacy of aPDT should become more cell-specific with minimization of induced phototoxicity to the host cell.

Photochemical and photophysical properties of cyclic tetrapyrroles, comprising the constitution of porphyrin-, chlorophyll-, chlorin-, and bacteriochlorin-derived structures, in serving as PSs have been extensively investigated [19,42,43]. To achieve the goal of these PSs toward working practically to inactivate bacteria and mammalian cells effectively, the agent needs to possess targeting accessibility while interacting with these cells. It is generally recognized that cationic surfactant molecules are able to disrupt the cell membrane of bacteria. This leads to biocidal effects by interacting with phospholipid components and anionic carboxylate moieties in the cytoplasmic membrane, thereby producing membrane distortion and protoplast lysis under osmotic stress [44,45]. In fact, polycationic polymers have emerged as promising substances for use as antimicrobial agents since the potential for resistance development is largely decreased [46]. Similarly, multiple positive charges attached to PSs can be utilized partially as a molecular strategy for targeting them to the bacterial cytoplasmic membrane to enhance antimicrobial effects by either themselves or in sophisticated formulations.

In terms of the charge factor and charge-density threshold for the optimal efficiency of biocidal action giving the loss of cell viability, they were investigated in combination of aPDI effects using functionalized photosensitizing chlorin derivatives and multicationic C_60_> monoduct or bisadduct conjugates [25,26,27,47,48,49,50]. In a part of these studies, we have demonstrated the aPDI activities upon the variation in the number of cationic charges per PS from 5 to 15 to confer biocidal properties. A group of multicationic PSs closely related to the present study were prepared with either 5, 10, or 15 discrete cationic charges as in Phe_Zn_-N_5_^+^, Chl_Zn_-N_10_^+^ (Zn^+2^ analogous of LC38^+^), or *m*Chl_Zn_-N_15_^+^, respectively [30]. The method used synthetically facile routes for attachment of either one, two, or three of well-defined pentacationic *N,N’,N,N,N,N*-hexapropyl-hexa(aminoethylene)amine-penta(quaternary methyl-ammonium iodide) arm [H_2_N(C_2_N^+^C_1_C_3_)_5_] to either [Zn^+2^]pheophorbide or [Zn^+2^]chlorin nucleus. Naturally occurring chlorophyll was applied as the starting compound in the synthesis with multicationic groups. This provided sufficient hydrophilicity for enhancing water solubility and, thus, converts the hydrophobic 18*π*-electron ring backbone of pheophytin a (Phe) to its amphiphilic equivalent.

Upon photoexcitation of these three reported PSs with either blue light (415 ± 15 nm) or red light (660 ± 15 nm), more bacterial killing with blue light as compared to the use of red light was found [30]. In the case of MRSA cells, they were very susceptible to light-mediated killing, particularly with Phe_Zn_-N_5_^+^ (five cationic charges) using a 415 nm light (20 J/cm^2^), giving complete eradication at a concentration as low as 100 nM. The compound of chlorin (Chl) analogous to Chl_Zn_-N_10_^+^ with ten cationic charges needs a light fluence of 20 J/cm^2^ at the concentration of 400 nM to produce the same eradication effect. Whereas the mesochlorin (*m*Chl) analogous *m*Chl_Zn_-N_15_^+^ having 15 cationic charges was found to give 5.0-log_10_ of killing slightly above full eradication. As a result, the order of inactivation effectiveness was summarized as Phe_Zn_-N_5_^+^ > Chl_Zn_-N_10_^+^ > *m*Chl_Zn_-N_15_^+^ [30]. Apparently, in all cases, covalently attached multiple (more than five) cationic charges on cyclic tetrapyrrole-derived PSs, giving appropriate amphiphilicity, were highly beneficial to provide photokilling ability of Gram-positive bacteria. In other examples of aPDI studies based on multicationic fullerenyl nano-PS derivatives using the same well-defined pentacationic H_2_N(C_2_N^+^C_1_C_3_)_5_ arm(s), decacationic C_60_> and C_70_> adducts with ten positive charges were found to exhibit much higher inactivation efficacy than those of the corresponding pentacationic analogous (five positive charges) [27]. Therefore, based on these analogous references, we selected the decacationic version of Chl_pd_-N_10_^+^ (LC38^+^ with ten positive charges, as dcPS, Figure 1) for this study, focusing on its targeting ability by electrostatic charges to anionic teichoic and terminal carboxylates of the outer peptidoglycan surface membrane layer of MRSA cells. In addition, two carbonyl amide [–(C=O)-NH–] moieties of both LC38^+^ and LC40e^+^ may form additional intermolecular H-bindings while interacting with teichoic carbonyl moieties of the Gram-positive bacterial cell wall [32,33]. Accordingly, the combination of these two types of interaction forces on the cell surface was proposed as the structural mechanism for enhancing targeting capability.

Furthermore, in our early studies, we focused exclusively on the role of cationic charges, which provide targeting ability through electrostatic interactions with negatively charged components of bacterial cell walls. Specifically, lipopolysaccharides bearing phosphate and carboxylic acid anions dominate the outer membrane of Gram-negative bacteria, while teichoic acids and terminal carboxylic acids contribute to the strong negative charge of the peptidoglycan layer in Gram-positive bacteria. These interactions guided the initial design of our mesochlorin conjugates. At a later stage, to move beyond purely static electrostatic interactions, we introduced antibiotics as synergistic targeting components to enhance aPDI through an alternative binding mechanism. The membrane-targeting vancomycin hydrochloride (VCM) molecule, as a cyclic glycopeptide antibiotic, was indicated as a drug of last resort traditionally for the treatment of serious, life-threatening infections by Gram-positive bacteria unresponsive to other antibiotics [51]. Its function is to block cell wall synthesis by binding to the C-terminal tripeptide (*L*-Lys-*D*-Ala-*D*-Ala) of the peptidoglycan precursor, offering a distinct and complementary mode of interaction compared with cationic charges [36]. While the mechanistic contributions of vancomycin and cationic charges could, in principle, be examined individually, such separate evaluations would not provide direct evidence of the synergistic effect resulting from their combination. As is often observed in synergistic systems, the combined effect can exceed the sum of the individual contributions. Therefore, we strategically advanced to evaluating conjugates that integrate both cationic charges and antibiotic moieties, allowing us to directly assess their combined targeting potential in enhancing aPDI efficacy. Hence, we extended the proposed structural modification, as described above, to a rational approach of formulating a combination design consisting of two pentacationic quaternary methyl-ammonium arm moieties and a VCM moiety in one covalent molecular conjugation, as a structure of VCMe-*m*Chl_pd_-N_10_^+^ (LC40e^+^, as Abx-dcPS, Figure 1), for performing synergistic targeting effects. The resulting conjugated compound LC40e^+^ consisting of a mesochlorin (*m*Chl) moiety and a VCM group covalently linked by an oligo(ethylene glycol) (EG_n_) chain, may serve as a new aPDI agent to exhibit the combinatory therapeutic effects of aPDI and antibiotics. The compound LC40e^+^ should also be capable of covalent co-delivery of both drug components concurrently to exhibit both bioactivities, while the previous analogous LC39^+^ (Figure 1) was found to be rather low or nearly inactive in either aPDI or antibiotic activities [30]. This large difference in molecular behavior supported our current structural modification from LC39^+^ to LC40e^+^ by taking the steric hindrance factor into consideration and separating mesochlorin and VCM moieties by a linear water-soluble EG_n_ chain. The EG_n_ extension allows either moiety to perform independently for collaborative effects, leading to significant improvement in bioresponsive functions.

In addition, as discussed above, the observed photocytotoxicity may originate from the facile production of ROS by either the chlorin ring moiety of LC38^+^ or the mesochlorin ring moiety of LC40e^+^ upon photoexcitation. The production efficiency depends highly on the prevention of hydrophobic planar chlorin or mesochlorin ring moieties from undergoing stacking aggregation due to *π*–*π* conjugation interactions in aqueous solution. Our design of two pentacationic quaternary methyl-ammonium iodide arms attaching to each ring significantly increases the steric hindrance around the 18-π-electron conjugation region and water solubility. Especially, the additional covalent bonding of a VCM with the EG_n_ linker should further reduce the tendency of LC40e^+^ molecules to stack on each other, which enhances the molecular separation and aPDI efficacy without triplet self-quenching issues. This, along with the enhanced targeting ability, may provide an explanation of the much higher aPDI activities of LC40e^+^ than those of LC37 and even LC38^+^.

## 4. Materials and Methods

### 4.1. Chemicals and Reagents

Reagents of trifluoroacetic acid (TFA), Pd(OAc)_2_·2H_2_O, methyl iodide (CH_3_I), *m*-chloroperoxybenzoic acid (*m*CPBA), ethylenediamine, pentaethylene hexamine, *n*-propionaldehyde, bis(3-aminopropyl) terminated oligo(ethylene glycol) (EG_n_, M_w_ 1500), poly(ethylene glycol) diglycidyl ether (M_n_ ~ 500), di(*n*-butyl)tin(IV) dilaurate (T12), trimethylamine, hydroiodic acid, and potassium carbonate were purchased from Sigma-Aldrich, Milwaukee, WI, USA, and used without further purification. Chlorella powder was purchased from Cellusyn Labs, LLC, Provo, UT, USA. Vancomycin hydrochloride (VCM) was purchased from Gold Biotechnology, St. Louis, MO, USA. Solvents were routinely distilled prior to use. The reagent *N,N’,N,N,N,N*-hexapropyl-penta(aminoethylene)amine (HPAA) was prepared by a modified method reported [28,38]. The singlet oxygen detecting probe ABMA was prepared previously in our report [28].

### 4.2. Synthesis of 15b-Methyl-13a,17c-di[aminopropylpolyoxyethylene]-chlorin e_6_ (Chl_pd_-EG_n_, LC37, nPS); 15b-Methyl-13a,17c-di[N,N’,N,N,N,N-hexapropyl-panta(aminoethylene)] Amide-[Pd^+2^]chlorin e_6_-deca(quaternary methyl-ammonium iodide) (Chl_Pd_-N_10_^+^, LC38^+^, dcPS) and 3a-Hydroxy-3b-aminoethylene-aminoPEGlycolated vancomycin-15b-methyl-13a,17c-di[N,N’,N,N,N,N-hexapropyl-panta(aminoethylene)]amide-mesochlorin-deca(quaternary methyl-ammonium iodide) (VCMe-mChl_Pd_-N_10_^+^, LC40e^+^, Abx-dcPS)

Preparation methods of LC37, LC38^+^, and LC40e^+^ follow those reported previously with slight modifications [30]. A brief synthetic procedure and the spectroscopic data were given in Appendix A.

### 4.3. Spectroscopic and Photophysical Characterization Instruments

Infrared spectra were recorded as KBr pellets on a Thermo Nicolet Avatar 370 FT-IR spectrometer (Thermo, Waltham, MA, USA). ^1^H NMR spectra were recorded on a Bruker Avance Spectrospin-500 spectrometer (Bruker, MA, USA). UV-vis or fluorescence spectra were recorded on a PerkinElmer Lambda 750 UV-vis-NIR spectrometer (PerkinElmer, CT, USA) or a PTI QuantaMaster^TM^ 40 fluorescence spectrofluorometer (PTI, NJ, USA), respectively. Light sources used in conjunction with spectroscopic measurements included a collimated white LED light with an output power of 2.0 W (Prizmatix, MI, USA), an OmniCure LX400 UV LED (PA, USA) spot curing lamp (operated at the emission peak maximum centered at 400 ± 5 nm with the peak irradiation intensity of 3.0 W/cm^2^, a focus lens of 6.0 mm in diameter, and a max power output of 320 mW), and a 650 nm laser pointer with a maximum power output of 200 mW for photoexcitation in the red wavelength. Transmission electron microscopy (TEM) measurements were carried out on a Philips EM400T transmission electron microscope (Thermo, Hillsboro, OR, USA). Confocal microscope measurements were carried out on a Leica TCS SP8 laser scanning (Leica Microsystems, Wetzlar, Germany) confocal microscope equipped with HyD GaAsP detectors and a DFC7000T color camera.

### 4.4. Light Sources Applied in aPDI Experiments

In in vitro experiments, two types of light sources were used for illumination of bacteria and mammalian cells. One was the Lumicare light source (Newport Beach, CA, USA), with the emission spectrum having *λ*_max_ centered at 660 nm and the full width at half maximum (FWHM) of 30 nm. It delivers a light spot in a diameter of 2.0 cm with an average irradiance of 40 mW/cm^2^. The other one was a light-emitting diode (LED) with a peak emission at 405 nm and FWHM of 25 nm (M405L2, Thorlabs, Newton, NJ, USA). It delivers a light spot in a diameter of 3.0 cm, giving an average irradiance of 24 mW/cm^2^. Irradiances of both light sources were measured by a power/energy meter (PM100D, Thorlabs). Light exposure (J/cm^2^) was calculated as the product of irradiance (W/cm^2^) and irradiation duration (sec).

### 4.5. Reactive Oxygen Species (ROS) Detection Using Singlet Oxygen (^1^O_2_)-Sensitive Fluorescent Probe

A synthetic compound, *α*,α’-(anthracene-9,10-diyl)bis(methylmalonic acid, tetrasodium salt) (ABMA), was used as a fluorescent probe for singlet oxygen (^1^O_2_) trapping experiments [30]. The quantity of ^1^O_2_ generated was monitored and counted by the relative intensity decrease in fluorescence emission of ABMA at 428 nm (*λ*_em_) under excitation wavelengths of 380 nm (*λ*_ex_). A typical probe solution was prepared by diluting a master solution of ABMA (2.0 mM in DMF, 10 µL) with 80-fold in volume of DMF (80 µL) mixed with 900-fold in volume of H_2_O (900 µL) in a cuvette (10 × 10 × 45 mm). The solution was added by a predefined volume of either LC37, LC38^+^, or LC40e^+^ in DMF (2.0 mM in DMF, 10 µL), followed by illumination periodically using either an ultrahigh power white-light LED lamp (Prizmatix, operated at the emission peak maxima centered at 451 and 530 nm with the collimated optical power output of >2.0 W in a diameter of 5.2 cm), a 400 nm LED curing light (320 mW), or a 650 nm laser (200 mW) for the generation of fluorescence emission spectra at different wavelength ranges. Progressive FL spectra were then collected on the PTI QuantaMaster^TM^ 40 fluorescence spectrofluorometer. Replicate measurements showed minimal variation.

### 4.6. Bacterial Stain and Culture Conditions

A clinical isolation of MRSA IQ0064 was studied as the model strain. Bacteria were routinely cultured on brain heart infusion (BHI) agar plates in a static incubator (Heracell, Heraeus, Newtown, CT, USA) under the condition of 37 °C and 5% CO_2_ or in BHI broth overnight in a C24 incubator shaker (New Brunswick Scientific, Edison, NJ, USA) at 34 °C and 150 rpm.

### 4.7. Human Keratinocytes and Culture Conditions

The human keratinocyte cell line HaCaT (kindly donated by Dr. Bin Zheng at Massachusetts General Hospital) was used as a representative host cell. HaCaT cells were cultured in Dulbecco’s modified Eagle’s medium (DMEM, Life Technologies, Grand Island, NY, USA) with 10% heat-inactivated fetal bovine serum, supplemented with penicillin (100 units/mL) and streptomycin (100 μg/mL) (Pen Strep, Life Technologies, Waltham, MA, USA), in a culture incubator (Heracell, Heraeus, MA, USA) at 37 °C under a 5% CO_2_-humidified atmosphere. HaCaT cells were used for the following experiments when 70~80% confluence was reached.

### 4.8. Planktonic MRSA Cell Preparation and Conditions for aPDI Evaluations

A similar methodology to that previously reported was applied for the evaluation [30]. Overnight bacterial cultures were centrifuged at 2500× *g* for 5.0 min, washed with PBS once, and adjusted in PBS to OD_600nm_  =  0.1, corresponding to a cell density of approximately 10^8^ CFU/mL for experimental use. Compounds of either dcPS or Abx-dcPS were diluted from the stock solution to define experimental concentrations by using PBS in the dark. For aPDI experiments, four study groups were included: (a) aPDI group, (PS+L+), in which MRSA suspensions were incubated with different concentrations of either dcPS or Abx-dcPS (v:v = 1:1) at 37 °C in the dark for 30 min, followed by light irradiation; (b) PS only group (PS+L–), in which MRSA cells were incubated with either dcPS or Abx-dcPS only but without light irradiation; (c) light only group, (PS–L+) in which MRSA suspensions were exposed to light irradiation alone without incubation with either dcPS or Abx-dcPS; and (d) untreated group (PS–L–); in which MRSA suspensions were without any intervention (no PS or L). The last three groups served as dark control, light control, and untreated control, respectively. Immediately after irradiation, bacterial suspensions were serially diluted by 10-fold and plated onto BHI agar plates. The plates were then incubated at 37 °C overnight. Viability of MRSA was measured using serial dilution and the CFU counting technique. All experiments were conducted in three independent replicates.

### 4.9. MRSA Biofilm Preparation and Conditions for aPDI Evaluations

MRSA cells were grown in BHI broth overnight. Cell density of bacterial suspensions was adjusted to approximately 10^6^ CFU/mL in BHI broth according to the OD_600nm_ value. Bacterial suspensions in 100-μL aliquots were inoculated in 96-well plates and incubated under 37 °C for 24 h to form mature biofilms. After incubation, MRSA biofilms were divided into four study groups in the same manner as those described in the study of planktonic bacteria above, i.e., the PS+L+, PS+L–, PS–L+, and PS–L– groups. Three technical replicates (three wells) were carried out in each group at each time. After incubating for 1.0 h with Abx-dcPS in PBS in the dark, the supernatant was removed and replaced with 100 μL fresh PBS. After subsequent light exposure, each well containing biofilm was scraped in PBS by using a 200 μL sterile pipette tip. Resulting bacterial cultures were transferred to 1.5 mL microcentrifuge tubes. This procedure was repeated once with fresh PBS (100 μL). Combined culture (200-μL) in each tube was then ultrasonicated (Bransonic M2800, Danbury, CT, USA) for 5.0 min to disperse the bacterial cells. These samples were subject to 10-fold serial dilution and plated on BHI agar plates for CFU counting. All experiments were performed in triplicate.

### 4.10. Examination of Photocytotoxicity to Human Cells

Immortalized HaCaT cells were used as representative host cells. To investigate the photocytotoxicity of aPDI to HaCaT cells, the cells were seeded in 96-well plates in a density of 5 × 10^4^ cells/well and incubated at 37 °C overnight. Following incubation, the adherent HaCaT cells were washed and then incubated for 30 min with either dcPS or Abx-dcPS at a defined concentration. The supernatant was then replaced with no-phenol Dulbecco’s modified Eagle medium (DMEM). The cells were irradiated by either 660 nm red light at an irradiance of 40 mW/cm^2^ or 405 nm blue light at an irradiance of 24 mW/cm^2^. At the end of light exposure, viability of the cells was determined by using the annexin V/propidium iodide (PI) apoptosis kit (Invitrogen, ThermoFisher Co., Waltham, MA, USA) and interpreted in terms of viable, apoptotic, or necrotic status. To this end, the irradiated cells were collected and washed once in cold Dulbecco’s phosphate-buffered saline (DPBS) and then re-suspended in 1× annexin-binding buffer. Aliquots (100 μL) of the cell suspension were incubated with 5.0 μL of FITC annexin V and 1.0 μL of PI (100 μg/mL) working solution at room temperature for 15 min. Immediately after the incubation, 1× annexin-binding buffer (400 μL) was added. The stained cells were analyzed using flow cytometry (Fortessa X-20, BD Biosciences, Franklin Lakes, NJ, USA), where fluorescence emissions at 530 and 610 nm were measured. These experiments were repeated in three independent replicates.

### 4.11. Statistical Analysis

Data were presented as the mean ± standard error. All statistical analyses were performed on SPSS 19.0 with their graphs plotted by Origin 9.1. Differences between different study groups were analyzed using a Kruskal–Wallis test. *p* < 0.05 was considered statistically significant.

### 4.12. Fixation of Bacteria and Host Cells for Targeting Imaging Studies

The same bacterial stain and culture conditions, as described above, were applied for both MRSA IQ0064 and HaCaT cells. Subsequently after incubation, either bacteria or host cell suspension was transferred to a centrifuge tube and centrifuged at 13,500× *g* for a period of 20–25 min. The resulting sentimental cell mass was washed gently three times by PBS. It was then added to a solution mixture of paraformaldehyde (1%) and glutaraldehyde (1.25%) to initiate and proceed with the imination reaction at 4.0 °C for 2.0 h that resulted in the fixation of cells. The preserved cells were centrifuged at 13,500× *g* for 10 min. After decanting off the liquid, the cells were washed three times with sodium cacodylate trihydrate buffer (0.1 M), followed by post-fixation with glutaraldehyde (2.0%) for 1.0 h. Subsequent centrifugation and the removal of excessive glutaraldehyde gave the stabilized cells, which were washed three times by sodium cacodylate buffer (0.1 M). Dehydration of either bacteria or host cells was later carried out by washings using acetone portions in a progressive increase in concentration from 30, 50, 70, 90, to 100%. Finally, the cells were suspended in anhydrous acetone as a stock solution for targeting imaging study measurements below.

### 4.13. Transmission Electron Microscopic (TEM) Image Evaluation

Transmission electron microscopy was performed to investigate morphological changes in bacteria cells. Both MRSA and host HaCaT cells, with or without pretreatment of either dcPS or Abx-dcPS, were prefixed with glutaraldehyde in the same manner as described above and suspended in acetone. A small portion of suspension solution (5.0 µL) was directly placed on a carbon−copper film grid, in a 200-mesh size, as the supporting plate. All solvents were removed via evaporation and further dried in vacuo for a period of 4.0 h prior to the TEM micrographic imaging data collection.

Ultra-thin section samples of MRSA cells were also prepared as bladed microtome slices, where the cells were embedded in Epon t812 (Tousimis, Rockville, MD, USA). They were cut using a Reichert-Jung Ultracut E microtome (Vienna, Austria) and collected on uncoated 200-mesh copper grids, followed by staining with uranyl acetate and lead citrate. During the measurement, multiple sections were microscopically analyzed. Images representing the most typical morphologies observed were presented in the study.

### 4.14. Confocal Microscopic Fluorescent Imaging Measurements

For the confocal microscopic fluorescent image investigation, the same glutaraldehyde-prefixed MRSA and host HaCaT cells, with or without pre-treatment of either dcPS or Abx-dcPS, in acetone suspension, as those used in TEM measurements, were applied. Prior to the cell casting, both the glass slide plate and cover glass were soaked in acetone and sonicated for 30 min three times. They were further rinsed by acetone three times to ensure a clean surface. A small portion of the cell suspension (5.0 µL in acetone) was placed on the glass slide, followed by removal of all solvents via evaporation and further dried in vacuo for a period of 4.0 h. A cover glass was then gently placed on the top of the glass slide to ensure the coverage of all sample cells. Clear nail polish liquid (Electron Microscopy Sciences, Hatfield, PA, USA) was subsequently applied cautiously on all edges between the cover glass and glass slide plate for sealing and binding them together. These cell sample plates were allowed to dry in air for 2.0 h.

In the case of the HaCaT cell sample preparation on slides, with or without pre-treatment of either dcPS or Abx-dcPS, the cells were cultured on a small round cover glass followed by staining with adenine–thymine binding 4′,6-diamidino-2-phenylindole (DAPI) as the fluorescent stain to the DNA nuclei of host cell. After washing away an excessive amount of DAPI by organic solvent, the cover glass was placed directly on a glass slide with the host cell side facing the glass slide. Clear nail polish liquid was subsequently applied cautiously on all edges between the cover glass and glass slide plates to seal and bind them together. These cell sample plates were allowed to dry in air for 2.0 h.

In the measurement, an oil-immersed objective lens (63×) was used throughout all experiments. One small drop of type F immersion liquid (*n*_e_^23^ = 1.5180, *V*_e_ = 46) was placed on the lens first; the premade sample was then placed on the observation stage with the cover glass side facing the objective lens. During the experiment, all image pictures were taken under the same condition. The irradiation source for photoexcitation was a 405 nm LED light with a consistent output power of 30 mW. The emission detector was set in a mode of two different wavelength channels with either a 450–550 nm bandwidth (channel 1) or a 640–780 nm bandwidth (channel 2). The shutter of light source and the gain of the detector were set at 2.0 and 1250, respectively. All images have a frame accumulation of four to ensure clear contrast against the black background. The raw images obtained from the confocal microscope were then used directly to generate the figures without any adjustments to contrast or brightness.

### 4.15. MIC Determination

To assess the potential toxicity of the VCM moiety in Abx-dcPS of VCMe-*m*Chl_pd_-N_10_^+^ (LC40e^+^) to MRSA cells (i.e., dark toxicity), the minimum inhibitory concentration (MIC) was determined by using the broth microdilution method [52]. The values were compared with those of vancomycin itself and VCM-*m*Chl_Zn_-N_5_^+^ (LC40^+^) [30]. The method used a traditional turbidity endpoint measurement on sterile 96-well microliter plates to generate a checker-board of 2.0-fold serial dilutions from column 1 (128 mg/mL) to column 11 (0.125 mg/mL) (column 12 = zero) for these compounds in 50% BHI broth. An aliquot of bacterial suspension (10 μL containing 10^4^ cells) was added to each well, and the plates were incubated at 37 °C for 15 h with vigorous shaking. The bacterial turbidity was monitored with a plate reader (SpectraMax M5 plate reader, Molecular Devices, Sunnyvale, CA, USA). These experiments were repeated three times.

## 5. Conclusions

A structure VCMe-*m*Chl_Pd_-N_10_^+^, consisting of a covalently bonded mesochlorin ring, two pentacationic quaternary methyl-ammonium iodide arms, and a VCM moiety with an EG_n_ linker, was designed and prepared by us following our previously published chemistry [30]. In this work, it was selected for the investigation of its targeting ability to MRSA cells and the subsequent aPDI activity. The method for decacationic functionalization on the chlorin moiety did not alter its basic *π*-conjugation system of the central ring, as evidenced by similar optical absorptions to those of other reported types of related chlorin derivatives [53,54,55]. Conversion of the chlorin ring to the corresponding mesochlorin moiety broadened the optical absorption over the full visible wavelength range in the case of Abx-dcPS, suitable for broadband white-light aPDI applications.

Evidently, the covalent application of ten positive charges and a Gram-positive bacteria-targeting VCM moiety in combination in an Abx-dcPS structure was capable of largely improving the targeting efficiency. Furthermore, by variation in the chain length of the EG_n_ linker of VCMe-*m*Chl_Pd_-N_10_^+^ to properly separate the mesochlorin ring moiety from the VCM moiety within the molecular structure, aPDI activity was found to be largely enhanced, which provided the nearly complete eradication (>6.5-log_10_ CFU reduction) of MRSA cells. These results were consistent with both targeting and the aPDI activity order of Abx-dcPS > dcPS >> nPS in structural relationship.

Even though the VCM moiety was applied for the targeting enhancement purpose at the initial stage of aPDI, in principle, it might serve as an effective antibiotic agent (MIC of ~2.0 µg/mL for LC40e^+^, similar to that of VCM itself) after aPDI to further kill residual MRSA cells at the same medication site and, consequently, prevent the reoccurrence of infection. Accordingly, the Abx-dcPS analogous LC40e^+^ should exhibit combinatory therapeutic effects of aPDI and antibiotics. It should also be able to provide covalent co-delivery of two drug components concurrently, leading to significant potential improvement in bioactive functions. Finally, the inherent photophysical properties of the chlorin scaffold, particularly its strong red fluorescence emission, provide the potential for real-time visualization and tracking of photosensitizer delivery.

## Figures and Tables

**Figure 1 antibiotics-14-00978-f001:**
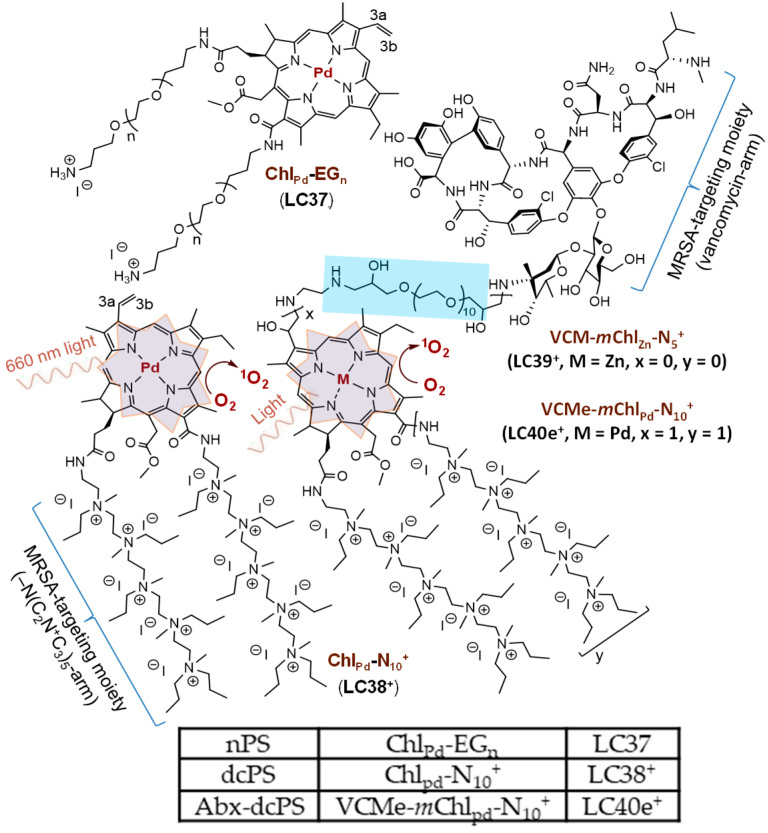
Chemical structures of three aPDT photosensitizing conjugates that were studied.

**Figure 2 antibiotics-14-00978-f002:**
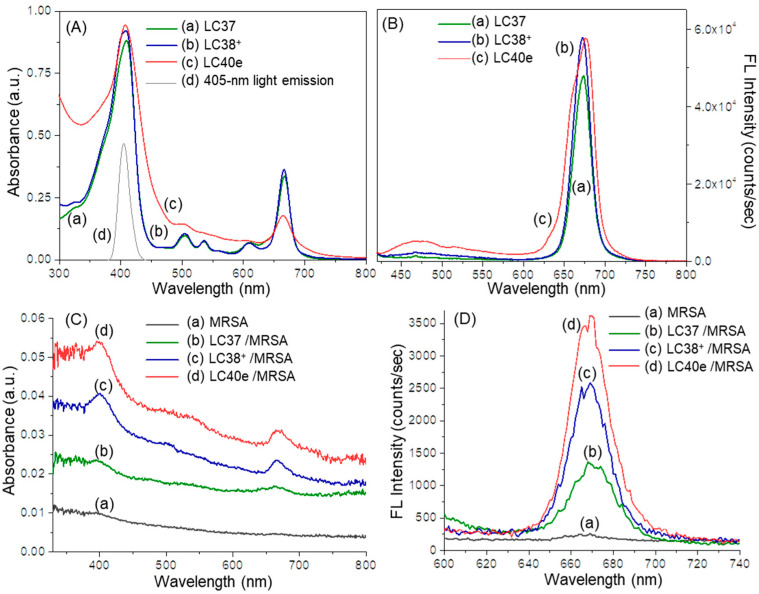
(**A**) UV-vis spectra of (a) LC37, (b) LC38^+^, (c) LC40e^+^, and (d) the emission spectrum of 405 nm LED light used in in vitro experiments and (**B**) fluorescence (FL) spectra (*λ*_ex_ 410 nm) of (a) LC37, (b) LC38^+^, and (c) LC40e^+^ in DMF at a concentration of 2.0 × 10^−5^ M. (**C**) Normalized UV-vis spectra of (a) MRSA alone and MRSA pretreated with (b) LC37, (c) LC38^+^, and (d) LC40e^+^ in the same concentration of 200 µM. (**D**) Normalized fluorescence emission spectra (*λ*_ex_ 410 nm) of (a) MRSA alone and MRSA pretreated (200 µM) with (b) LC37, (c) LC38^+^, and (d) LC40e^+^. Both (**C**,**D**) were taken in a solution of anhydrous acetone in a concentration as prepared after fixation with paraformaldehyde and glutaraldehyde. Normalization calculation was performed using the measured extinction coefficient (*ε*) value to an effective number of dcPS or Abx-dcPS molecules and MRSA cell counts per mL obtained by flow cytometric analysis.

**Figure 3 antibiotics-14-00978-f003:**
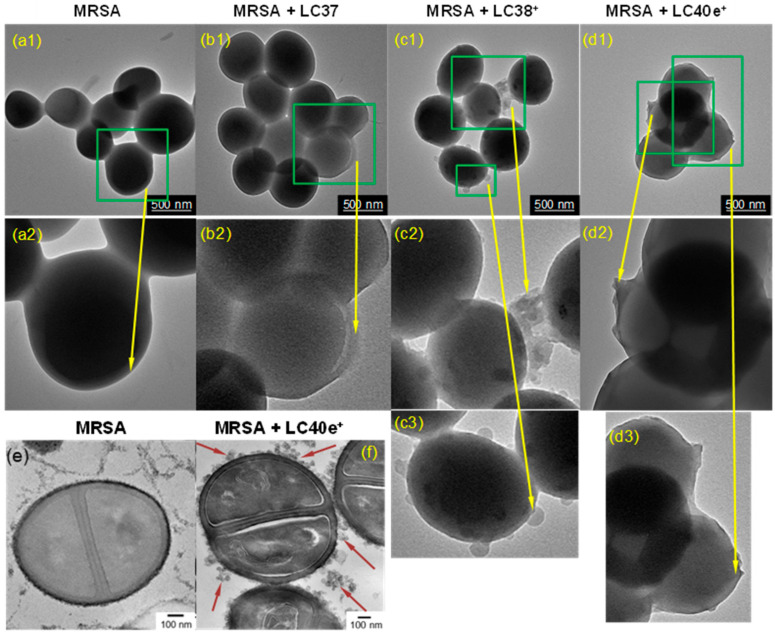
TEM images of (**a1**,**a2**) MRSA cells alone and MRSA cells pretreated with (**b1**,**b2**) LC37, (**c1**–**c3**) LC38^+^, and (**d1**–**d3**) LC40e^+^ in a concentration of 200 µM, where two or three different magnifications were displayed to show the extracellular binding of nPS (**b1**,**b2**), dcPS (**c1**–**c3**), or Abx-dcPS (**d1**–**d3**). (**e**) A bladed microtome slice section of an MRSA sample in the Epon t812 polymer matrix for comparison of morphological changes with those of (**f**) the similar slice section of MRSA pretreated with LC40e^+^ (200 µM), showing many physically attached Abx-dcPS at different locations, as marked by red arrows, clearly differentiable from (**e**).

**Figure 4 antibiotics-14-00978-f004:**
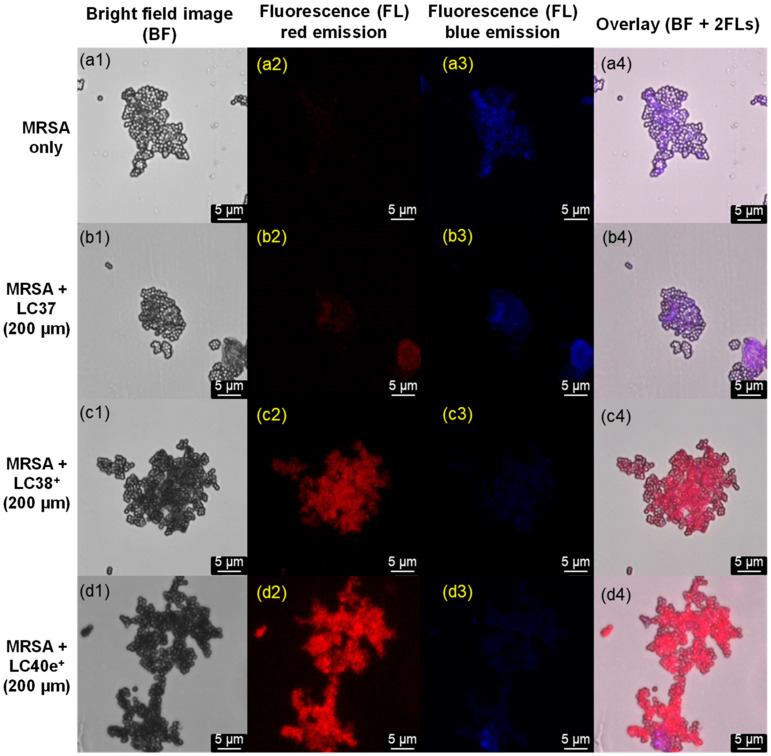
Confocal microscopic images of (**a1**–**a4**) MRSA cells alone and MRSA cells pretreated with (**b1**–**b4**) LC37, (**c1**–**c4**) LC38^+^, and (**d1**–**d4**) LC40e^+^ in a concentration of 200 µM, where four separated columns were displayed to show the images under either bright field (left), FL red emission (640–780 nm, middle left), FL blue emission (450–550 nm, middle right), or overlay of all three fields in the last column on the right. In all cases, the irradiation excitation at 405 nm was applied.

**Figure 5 antibiotics-14-00978-f005:**
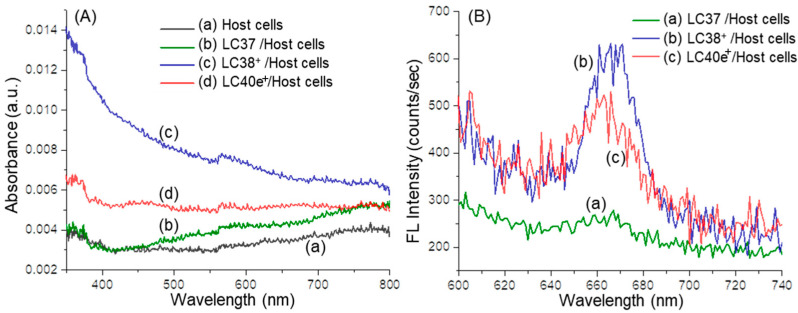
(**A**) Normalized UV-vis spectra of (a) host cells alone and host cells pretreated with (b) LC37, (c) LC38^+^, and (d) LC40e^+^ in the same concentration of 200 µM. (**B**) Normalized fluorescence emission spectra (*λ*_ex_ 410 nm) of host cells pretreated (200 µM) with (a) LC37, (b) LC38^+^, and (c) LC40e^+^. Both (**A**,**B**) were taken in a solution of anhydrous acetone in a concentration as prepared after fixation with paraformaldehyde and glutaraldehyde. Normalization calculation was performed using the measured extinction coefficient (*ε*) value to an effective number of dcPS or Abx-dcPS molecules and host cell counts per mL obtained by flow cytometric analysis.

**Figure 6 antibiotics-14-00978-f006:**
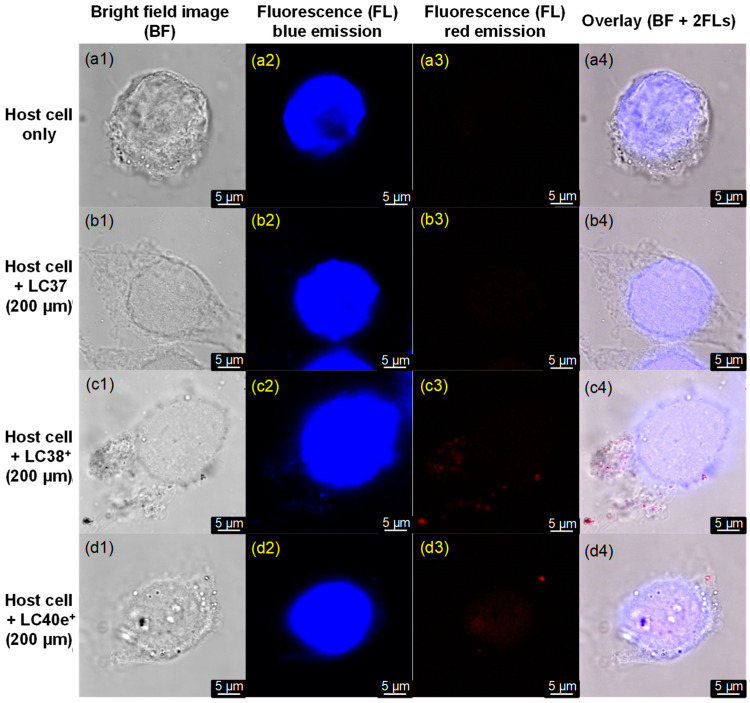
Confocal microscopic images of (**a1**–**a4**) host cells alone and host cells pretreated with (**b1**–**b4**) LC37, (**c1**–**c4**) LC38^+^, and (**d1**–**d4**) LC40e^+^ in a concentration of 200 µM, where four separated columns were displayed to show the images under either bright field (left), blue FL emission (450–550 nm, middle left), red FL emission (640–780 nm, middle right), or an overlay of all three fields in the last column on the right. In all cases, the irradiation excitation at 405 nm was applied.

**Figure 7 antibiotics-14-00978-f007:**
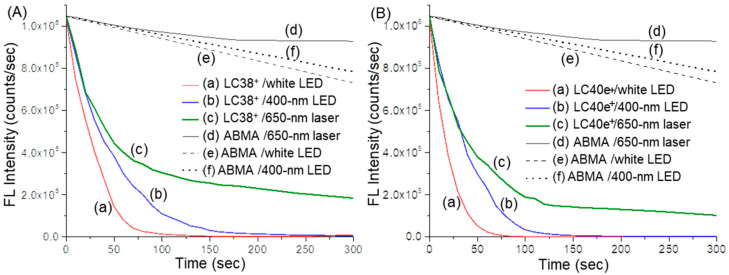
Time-dependent fluorescent emission intensity of ABMA, as the synthetic ^1^O_2_ trapping probe with the photoexcitation at *λ*_ex_ 380 nm and emission at *λ*_em_ 428 nm, using the aPDI agent of (**A**) LC38^+^ and (**B**) LC40e^+^, in a concentration of 20 µM (DMF–H_2_O/1:9), under irradiation of (a) white LED (2.0 W), (b) 400 nm LED (320 mW), and (c) 650 nm laser (200 mW) light. ABMA was also applied alone under the same irradiation condition using (d) 650 nm laser, (e) white LED, and (f) 400 nm LED light as the background reference for comparison.

**Figure 8 antibiotics-14-00978-f008:**
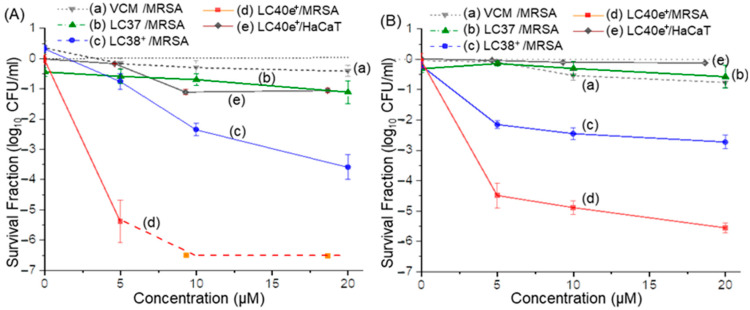
In vitro aPDI activities of (a) vancomycin alone, (b) LC37, (c) LC38^+^, and (d) LC40e^+^ on planktonic MRSA IQ0064 cells under the dcPS or Abx-dcPS concentrations of up to 20 µM. Irradiation was carried out with (**A**) the radiant exposure of 115 J/cm^2^ at 660 nm (LED light, 40 mW/cm^2^) and (**B**) the radiant exposure of 20 J/cm^2^ at 405 nm (LED light, 24 mW/cm^2^). All experiments were conducted in three independent replicates. All data were represented as log_10_ survival fractions as a function of aPDI agent concentration, where the dashed red line in A(d) represents the detection limit of MRSA CFU. Human epithelial keratinocyte HaCaT cells were also included in the study, under the same experimental conditions, for the study of photocytotoxicity to human cells with the data in (e).

**Figure 9 antibiotics-14-00978-f009:**
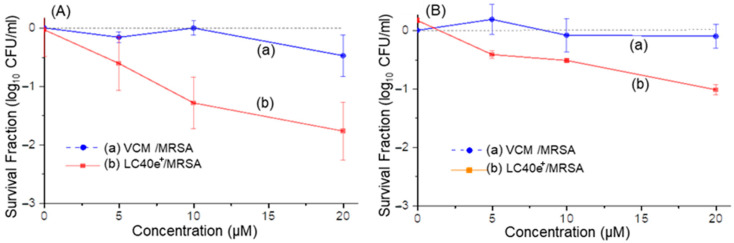
In vitro aPDI activities of LC40e^+^ on the 24 h-old biofilms of MRSA IQ0064 using a concentration of Abx-dcPS up to 20 µM, (a) dark control and (b) aPDI group. Irradiation was carried out with (**A**) the radiant exposure of 115 J/cm^2^ for 660 nm LED light (40 mW/cm^2^) and (**B**) the radiant exposure of 20 J/cm^2^ for 405 nm LED light (24 mW/cm^2^). All experiments were conducted in three independent replicates. All data were represented as log_10_ survival fractions as a function of both PS concentrations.

## Data Availability

The data supporting the findings of this study are available from the corresponding authors upon reasonable request.

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
