# Peer review of "Selective Targeting and Enhanced Photodynamic Inactivation of Methicillin-Resistant *Staphylococcus aureus* (MRSA) by a Decacationic Vancomycin–Mesochlorin Conjugate"

_antibiotics, 2025, doi:10.3390/antibiotics14100978_

Round 1

Reviewer 1 Report

Comments and Suggestions for Authors

The authors compared the photosensitizer (nPS) Chlpd-EGn (LC37) with decacationic photosensitizer (dcPS) Chlpd-N10+ (LC38+) and antibiotic decacationic photosensitizer (Abx-dcPS) VCMe-mChlpd-N10+ (LC40e+). The novelty of this study is in the use of oligo(ethylene glycol) linker in the making of conjugate. The resulting molecules were tested against MRSA as well as host cell HaCaT in vitro. A photosensitizing alone is an antibiotic when activated by the corresponding visible light. On the other hand, vancomycin is an antibiotic targetting the cell wall. Conjugates containing of both compounds may result in the molecule with more active binding site, thus increasing the antibacterial efficacy against resistant pathogens.

The results showed that nPS, dcPS, and Abx-dcPS inhibit the growth of Gram -positive bacteria (MRSA), but not host cell. Moreover, characterization of the behaviour from the three samples are well presented in the different figures. Activation of PDI was observed on different light wavelength. However, there is one question whether nPS and dcPS have antibiofilm activity against MRSA that can be added to figure 9.  

Minor:

  • Please use the citation of the reference in the end of the sentence.
  • Please be consistent in the writing style of the conjugates name in Figure 1 with the name in the text, especially for the “Pd”.
  • Figure 3 e and f need to be in the same magnification
  • Line 231 and 674 Ultrathin --> ultra-thin

Author Response

The authors compared the photosensitizer (nPS) Chlpd-EGn (LC37) with decacationic photosensitizer (dcPS) Chlpd-N10+ (LC38+) and antibiotic decacationic photosensitizer (Abx-dcPS) VCMe-mChlpd-N10+ (LC40e+). The novelty of this study is in the use of oligo(ethylene glycol) linker in the making of conjugate. The resulting molecules were tested against MRSA as well as host cell HaCaT in vitro. A photosensitizing alone is an antibiotic when activated by the corresponding visible light. On the other hand, vancomycin is an antibiotic targetting the cell wall. Conjugates containing of both compounds may result in the molecule with more active binding site, thus increasing the antibacterial efficacy against resistant pathogens.

We sincerely appreciate your positive and encouraging evaluation of our manuscript. Thank you for your kind support and constructive input throughout the review process.

The results showed that nPS, dcPS, and Abx-dcPS inhibit the growth of Gram -positive bacteria (MRSA), but not host cell. Moreover, characterization of the behaviour from the three samples are well presented in the different figures. Activation of PDI was observed on different light wavelength. However, there is one question whether nPS and dcPS have antibiofilm activity against MRSA that can be added to figure 9.  

We sincerely appreciate your suggestion of including biofilm experiments in Figure 9. Initially, we conducted experiments on planktonic MRSA and observed that only LC40e+ exhibited promising inactivation, whereas LC37 and LC38+ showed relatively low activity, which was consistent with our expectations. Since LC37 and LC38+ did not demonstrate satisfactory performance against planktonic MRSA, we did not pursue further testing with these compounds on biofilms.

A clarification and explanation have been added to the manuscript in Section 2.6.:

“Based on the results of aPDI efficiency against planktonic MRSA shown in Figure 8, LC37 and LC38+ did not display promising activity due to the lack of selective targeting antennas, in contrast to LC40e+. Therefore, only LC40e+ was selected for evaluation of its aPDI efficiency on biofilms.”

Minor:

  • Please use the citation of the reference in the end of the sentence.

    We appreciate your suggestion. All references have been carefully reviewed, and the citations have been moved to the end of the sentence.

  • Please be consistent in the writing style of the conjugates name in Figure 1 with the name in the text, especially for the “Pd”.

    The names in figure 1 have been reformatted to have the same writing style.

  • Figure 3 e and f need to be in the same magnification

    Figure 3e has now been replaced by another image with the same magnification.

  • Line 231 and 674 Ultrathin --> ultra-thin

    The typo “ultrathin” has been changed to “ultra-thin”.

Reviewer 2 Report

Comments and Suggestions for Authors

The manuscript entitled “Selective Targeting and Enhanced Photodynamic Inactivation of Methicillin-Resistant Staphylococcus aureus (MRSA) by a Decacationic Vancomycin–Mesochlorin Conjugate” presents interesting findings on a novel conjugate with potential application against MRSA. While the study provides valuable knowledge, several issues need to be addressed.

Comments

  1. The manuscript introduces a vancomycin–mesochlorin conjugate with decacationic arms for enhanced MRSA targeting. While the approach is interesting, the authors should emphasize more clearly how this design differs from previously reported vancomycin–photosensitizer conjugates. A comparative discussion with earlier vancomycin–porphyrin or chlorin conjugates would strengthen the novelty claim.
  2. The study demonstrates improved MRSA binding and photodynamic inactivation (aPDI) efficacy, but the mechanistic contribution of vancomycin versus cationic charge is not fully separated.
  3. The biofilm results (only ~2-log reduction vs. >6.5-log in planktonic cells) are relatively modest. Authors should discuss strategies to overcome biofilm resistance, such as penetration studies (e.g., CLSM z-stack imaging, EPS disruption analysis) or combination with dispersal agents. (optional)
  4. Some figures (e.g., CFU reduction graphs, ROS generation kinetics) lack sufficient statistical detail (number of replicates, error bars, statistical tests applied). The manuscript would benefit from a more rigorous presentation of quantitative data.
  5. The manuscript alternates between “aPDT” and “aPDI.” Please define clearly at the beginning and use consistently throughout.
  6. Several sentences are complex and difficult to follow (e.g., Introduction lines 52–61). Editing for conciseness and clarity is recommended.
Comments on the Quality of English Language

The English could be improved to more clearly express the research.

Author Response

The manuscript entitled “Selective Targeting and Enhanced Photodynamic Inactivation of Methicillin-Resistant Staphylococcus aureus (MRSA) by a Decacationic Vancomycin–Mesochlorin Conjugate” presents interesting findings on a novel conjugate with potential application against MRSA. While the study provides valuable knowledge, several issues need to be addressed.

We sincerely appreciate your positive and encouraging evaluation of our manuscript. Thank you for your kind support and constructive input throughout the review process.

Comments

  1. The manuscript introduces a vancomycin–mesochlorin conjugate with decacationic arms for enhanced MRSA targeting. While the approach is interesting, the authors should emphasize more clearly how this design differs from previously reported vancomycin–photosensitizer conjugates. A comparative discussion with earlier vancomycin–porphyrin or chlorin conjugates would strengthen the novelty claim.

    We appreciate your suggestion comparing our current results with our earlier study. In that previous work, our focus was primarily on investigating how variations in the number of charges on mesochlorin influenced activity. We subsequently introduced vancomycin (VCM) as a synergistic targeting component for enhancing aPDI. However, due to the large steric hindrance of VCM, its accessibility to the MRSA cell surface was greatly reduced, resulting in minimal aPDT efficacy. Building on these findings, in the present manuscript we designed and studied a combinatorial aPDT–antibiotic compound, LC40e+, a molecule incorporating 10 charges, mesochlorin, a short polymer linker, and vancomycin, along with new biological data demonstrating its selective binding and potent antimicrobial photodynamic effects against MRSA.

    A paragraph has been added to the manuscript in Section 2.1. to address this matter:

    “The development of LC40e+ is based on an evolutionary series of mesochlorin conjugates. Our earlier research demonstrated that varying the number of positive charges (from 2 to 15) on the mesochlorin scaffold significantly impacts antibacterial activity, with 10 charges yielding the highest efficacy [39]. Building on these findings, vancomycin (VCM) was introduced as a synergistic targeting component to enhance aPDI. However, our recent study revealed that when the linkage between the mesochlorin core and VCM was shorter than four C–C or C–N bonds, the bulky VCM moiety caused substantial steric hindrance, greatly reducing accessibility to the MRSA cell surface and resulting in minimal aPDT efficacy. To overcome this limitation, we revised the design by introducing an EG10 linker using identical synthetic procedures. This modification successfully alleviated steric hindrance. Consequently, we prepared LC40e+, a combinatorial aPDT–antibiotic compound consisting of two covalently bonded quaternary ammonium pentacationic arms on the mesochlorin chromophore core and a short polymer linker conjugated to VCM, for evaluation of antibacterial photodynamic inactivation (aPDI) activity.”

  2. The study demonstrates improved MRSA binding and photodynamic inactivation (aPDI) efficacy, but the mechanistic contribution of vancomycin versus cationic charge is not fully separated.

    We fully understand your concerns. In our early studies, we evaluated only cationic charges, which target the negatively charged components of bacterial cell walls. At a later stage, we introduced antibiotics as synergistic targeting components to enhance aPDI through an alternative binding mechanism. While the contributions of vancomycin and cationic charges could be studied separately, such tests would not directly demonstrate their synergistic effect. Therefore, we focused on evaluating the combined strategy of charges and antibiotic conjugation.

    A discussion has been added to the manuscript in Discussion section to address this matter:

    “Furthermore, in our early studies, we focused exclusively on the role of cationic charges, which provide targeting ability through electrostatic interactions with negatively charged components of bacterial cell walls. Specifically, lipopolysaccharides bearing phosphate and carboxylic acid anions dominate the outer membrane of Gram-negative bacteria, while teichoic acids and terminal carboxylic acids contribute to the strong negative charge of the peptidoglycan layer in Gram-positive bacteria. These interactions guided the initial design of our mesochlorin conjugates. At a later stage, to move beyond purely static electrostatic interactions, we introduced antibiotics as synergistic targeting components to enhance aPDI through an alternative binding mechanism. Membrane-targeting vancomycin hydrochloride (VCM) molecule, as a cyclic glycopeptide antibiotic, was indicated as a drug of last resort traditionally for the treatment of serious, life-threatening infections by Gram-positive bacteria unresponsive to other antibiotics [55]. Its function is to block cell wall synthesis by binding to the C-terminal tripeptide (L-Lys-D-Ala-D-Ala) of the peptidoglycan precursor, offering a distinct and complementary mode of interaction compared with cationic charges [40, 56]. While the mechanistic contributions of vancomycin and cationic charges could, in principle, be examined individually, such separate evaluations would not provide direct evidence of the synergistic effect resulting from their combination. As is often observed in synergistic systems, the combined effect can exceed the sum of the individual contributions. Therefore, we strategically advanced to evaluating conjugates that integrate both cationic charges and antibiotic moieties, allowing us to directly assess their combined targeting potential in enhancing aPDI efficacy.”

  3. The biofilm results (only ~2-log reduction vs. >6.5-log in planktonic cells) are relatively modest. Authors should discuss strategies to overcome biofilm resistance, such as penetration studies (e.g., CLSM z-stack imaging, EPS disruption analysis) or combination with dispersal agents. (optional)

    We appreciate the reviewer’s insightful comment regarding the relatively modest biofilm results. This observation is indeed consistent with the well-documented resistance of biofilms, largely attributed to the protective EPS matrix and limited penetration of therapeutic agents. In this study, although our primary aim was to establish proof-of-concept efficacy of LC40e+ against biofilm-associated MRSA, we still performed dose-dependent experiments, which higher inactivation of biofilms can be achieved at higher aPDI doses. However, we fully agree that additional strategies could further enhance activity. For example, future work will include penetration studies using CLSM z-stack imaging to directly visualize distribution within biofilms, EPS disruption analysis to evaluate matrix destabilization, and combination approaches with dispersal agents to facilitate deeper penetration and improved bacterial inactivation.

    A note discussing these strategies has been added to the revised manuscript in Section 2.6.:

    “Although LC40e+ achieved significant activity against biofilm-associated MRSA (~2-log reduction), its efficacy seemed to be lower compared with planktonic cells (>6.5-log reduction). It should be noted that in a dose-dependent experiment, higher inactivation of biofilms can be achieved by applying higher aPDI doses. This outcome is consistent with the inherent resistance of biofilms due to their dense extracellular polymeric substance (EPS) matrix and restricted photosensitizer penetration. In future research, particular attention will be given to strategies to enhance biofilm eradication, including penetration analysis by CLSM z-stack imaging, EPS disruption assays to assess matrix destabilization, and combination approaches with biofilm-dispersal agents.”

  4. Some figures (e.g., CFU reduction graphs, ROS generation kinetics) lack sufficient statistical detail (number of replicates, error bars, statistical tests applied). The manuscript would benefit from a more rigorous presentation of quantitative data.

    We appreciate your suggestion. The error bars in Figures 8 and 9 represent variability across the data. All experiments were conducted in three independent replicates. Statistical analysis is described in Section 4.11. For ROS generation evaluation, detailed methodology is provided in Section 4.5, and replicate measurements showed minimal variation. We have now included this information briefly in the figure legends to facilitate easier tracking by readers.

  5. The manuscript alternates between “aPDT” and “aPDI.” Please define clearly at the beginning and use consistently throughout.

    We fully understand your concerns. aPDT stands for antibacterial photodynamic therapy, which refers to the practical therapeutic application of the photosensitizers. In contrast, aPDI stands for antibacterial photodynamic inactivation, which specifically refers to the antibacterial inactivation experiments conducted with these photosensitizers. Both terms were clearly defined in Abstract where they first appeared in the manuscript. Their usage has been checked for consistency throughout the manuscript.

  6. Several sentences are complex and difficult to follow (e.g., Introduction lines 52–61). Editing for conciseness and clarity is recommended.

    We appreciate your suggestion. Introduction (lines 52–61) has now been rewritten for improved clarity and readability:

    “Methicillin-resistant Staphylococcus aureus (MRSA) strains are capable of producing various toxins and enzyme proteins, which contribute to their high pathogenicity and make them potentially lethal to humans [1-4]. The persistent presence of these bacterial surface-binding factors on host cells can lead to a range of severe infections, particularly skin and soft tissue infections, and potentially others. The continued emergence of MRSA variants, which adapt to different community settings and exhibit varying levels of resistance to conventional antibiotics, presents significant therapeutic challenges [5-7]. Although new antibacterial drugs are continually being developed, multidrug-resistant strains continue to arise, highlighting the urgent need to explore alternative therapeutic strategies [8-10].”

Reviewer 3 Report

Comments and Suggestions for Authors

The research focuses on investigating the antibacterial activity and photogeneration of singlet oxygen using mesochlorin-based photosensitizers. A positive effect was observed from the covalent combination of ten positive charges and a vancomycin moiety targeting MRSA, leading to the development of an effective antibacterial agent. Furthermore, an improvement in MRSA cell-targeting efficiency was demonstrated. However, several points need clarification prior to the publication of this article:

  1. Introduction Section:
    Photosensitizers (PSs) rely on molecular oxygen (O₂) as a substrate to produce reactive oxygen species (ROS). Two major photochemical mechanisms are typically considered:

    • Type I mechanism, involving electron transfer from the photosensitizer to O₂ to form a superoxide anion radical (O₂⁻•),

    • Type II mechanism, involving energy transfer from the excited triplet state of the PS to triplet oxygen, generating singlet oxygen (¹O₂).
      The manuscript focuses on the Type II mechanism. Could LC37 or the other PSs presented in this article also generate superoxide anion radicals via a Type I mechanism?

  2. Figures 3e and 3f (Page 7). The magnifications used for the images are different. For consistent and unambiguous detection of LC40e⁺ conjugates, the images should be compared at the same magnification.

  3. Figure 4. Was the excitation laser power the same across for the confocal imaging of MRSA cells pretreated with LC37, LC38⁺, and LC40e⁺?

  4. Figure 6. It seems that the images showing blue and red fluorescence emissions are mixed up. Please verify and correct this if necessary.

  5. The manuscript does not include any information about singlet oxygen generation by LC37. This data should be added or clarified.

  6. The manuscript contains numerous abbreviations for photosensitizers, such as nPS (LC37), dcPS (LC38⁺), and Abx-dcPS (LC40e⁺). It would be helpful to include a table summarizing all investigated PSs, along with their structures and designations, for the reader's convenience.

Author Response

The research focuses on investigating the antibacterial activity and photogeneration of singlet oxygen using mesochlorin-based photosensitizers. A positive effect was observed from the covalent combination of ten positive charges and a vancomycin moiety targeting MRSA, leading to the development of an effective antibacterial agent. Furthermore, an improvement in MRSA cell-targeting efficiency was demonstrated. However, several points need clarification prior to the publication of this article:

  1. Introduction Section:
    Photosensitizers (PSs) rely on molecular oxygen (O₂) as a substrate to produce reactive oxygen species (ROS). Two major photochemical mechanisms are typically considered:

    • Type I mechanism, involving electron transfer from the photosensitizer to O₂ to form a superoxide anion radical (O₂⁻•),

    • Type II mechanism, involving energy transfer from the excited triplet state of the PS to triplet oxygen, generating singlet oxygen (¹O₂).
      The manuscript focuses on the Type II mechanism. Could LC37 or the other PSs presented in this article also generate superoxide anion radicals via a Type I mechanism?

      We thank the reviewer for raising this insightful question. We had the same curiosity and therefore evaluated the potential Type I mechanism for these photosensitizers. However, we did not observe significant Type I photochemistry for these compounds.

      A paragraph has been added to the manuscript in Section 2.4. to discuss this matter:

      “We previously reported the possibility to achieve intramolecular photoinduced electron-transfer from counter iodide anions to the photoexcited fullerene cage moiety of water-soluble C60[>M(C3N6+C3)2]-(I)10 leading to, eventually, the formation of O2· [39]. It was also previously reported that the iodide anion (I) is capable of initiating a reaction with 1O2 to form several types of ROS, including O2·, H2O2, and HO· [40]. The reaction was associated with a plausible mechanism of electron-transfer event from I to 1O2 that led to the possible formation of O2·. Accordingly, we performed similar experiments to monitor the production of O2· by the same series of mcPSs to address whether it is the instance. In the O2·-trapping reaction, a regioisomeric mixture of superoxide radical-reactive fluorescent probe, potassium bis(2,4-dinitrobenzenesulfonyl)-2’,4’,5’,7’-tetrafluorofluorescein-10’(or 11’)-carboxylate isomers (DNBs-TFFC) were used for the experiment. High reaction selectivity of this probe toward O2· was reported with a O2·/1O2 sensitivity ratio of 46 [41]. However, all compounds we selected for evaluation induced roughly similar intensity of fluorescence from the probe with the fluctuation of only ±1% from that of blank control. Therefore, we concluded that no significant production of O2· was observed regardless of the variation of mcPSs.”

  2. Figures 3e and 3f (Page 7). The magnifications used for the images are different. For consistent and unambiguous detection of LC40e⁺ conjugates, the images should be compared at the same magnification.

    Thank you for pointing out the concern. Figure 3e has now been replaced by another image with the same magnification.

  3. Figure 4. Was the excitation laser power the same across for the confocal imaging of MRSA cells pretreated with LC37, LC38⁺, and LC40e⁺?

    The experimental condition of confocal microscopy for MRSA is identical across the evaluation. Same excitation fluence at 405 nm was applied to acquire the image. Raw images captured by the confocal microscope was attached. All figures presented in the manuscript were generated directly from these raw data without any modifications to contrast or brightness.

    A clarification has been added to manuscript in Section 4.14. to address this matter:

    “During the experiment, all image pictures were taken under the same condition. Irradiation source for photoexcitation was a 405-nm LED light with a consistent output power of 30 mW. The emission detector was set in a mode of two different wavelength channels with either a 450–550-nm band width (channel 1) or a 640–780-nm band width (channel 2). The shutter of light source and the gain of detector were set at 2.0 and 1250, respectively. All images have a frame accumulation of four to ensure clear contrast against the black background. The raw images obtained from the confocal microscope were then used directly to generate the figures without any adjustments to contrast or brightness.”

  4. Figure 6. It seems that the images showing blue and red fluorescence emissions are mixed up. Please verify and correct this if necessary.

    We thank the reviewer for pointing out the mistake. The figure notations in figure 6 have been corrected.

  5. The manuscript does not include any information about singlet oxygen generation by LC37. This data should be added or clarified.

    We thank the reviewer for raising this question. LC37 was only serving as the reference compound for comparison of targeting ability. LC37 gives negligible targeting toward the cells. Therefore, regardless of ROS generation, such minimal targeting would not result in meaningful aPDI efficiency. Thus, its ROS evaluation was not performed.

    A clarification has been added to the manuscript in Section 2.4. to address this matter:

    “The neutral LC37 was included only as a reference for comparing targeting ability. It demonstrated negligible targeting toward MRSA. Since effective aPDI requires not only ROS generation but also sufficient photosensitizer localization at the target site, compounds with minimal targeting ability are unlikely to achieve meaningful inactivation, regardless of their ROS yield. Therefore, further evaluation of ROS generation by LC37 was not pursued, as its lack of targeting rendered it unsuitable for effective aPDI.therefore, ROS evaluation was not conducted for this compound.”

  6. The manuscript contains numerous abbreviations for photosensitizers, such as nPS (LC37), dcPS (LC38⁺), and Abx-dcPS (LC40e⁺). It would be helpful to include a table summarizing all investigated PSs, along with their structures and designations, for the reader's convenience.

    We thank the reviewer for the suggestion, a table that summarizes the compounds was added to figure 1.

Reviewer 4 Report

Comments and Suggestions for Authors

This study investigated the enhanced binding and targeting ability of a compound called VCMe-mChlPd-N10+ (LC40e+) for antimicrobial photodynamic inactivation (aPDI) activity on the surface of Gram-positive bacterium MRSA cells. MRSA biofilm efficacy was found to be lower in the biofilm membrane than in planktonic cells. The authors conducted a very detailed study. Here are my few suggestions that could help I nth e improvement of the manuscript:

  1. Can there be a comparative study between different Gram-positive microbes to study aPDI efficacy? What was the reason to solely focus on mRSA?
  2. Also, a study was conducted with a single antibiotic. What was the rationale behind the single antibiotic test?
  3. There are few limitations already mentioned by authors about TEM imaging. Apart from TEM imaging, what other methods were used to find the mechanism of Abx-dcPS penetration?
  4. Concentration of Abx-dcPs id too high 10-20μm. What is the range of optimising and balancing the safety and efficiency of the treatment?
  5. aPDi activity on mRSA biofilms showed higher antimicrobial activity than planktonic cells. Is there any alternate strategy that has been tried to increase the efficiency of biofilm penetration?
  6. The non-linear graph showed a gradual decrease in ABMA concentration in the solution. Are there any precautions used by the authors?
  7. The material and method section like 4.8 lacks citations. Apart from a detailed description, it is advisable to add a citation for every methodology used.
  8. In section 2.4 ABMA is previously synthesized. Is the synthesis method of the ABMA compound used for ROS is described in this study? There are various literature cited with 3 self-citations. Does it point to the synthesis of direct sources of compounds or previously synthesized ABMA was used?
  9. What are the main challenges raised in this study, and why did the VCMe-mChlPd-N10+ (LC40e+) conjugate combine vancomycin with a decacation strategy to target and inactivate methicillin-resistant Staphylococcus aureus (MRSA) selectively? Furthermore, what important advantages does this combination approach offer compared to conventional antibiotics or isolated antimicrobial photodynamic inactivation (aPDI) treatments?

Author Response

This study investigated the enhanced binding and targeting ability of a compound called VCMe-mChlPd-N10+ (LC40e+) for antimicrobial photodynamic inactivation (aPDI) activity on the surface of Gram-positive bacterium MRSA cells. MRSA biofilm efficacy was found to be lower in the biofilm membrane than in planktonic cells. The authors conducted a very detailed study. Here are my few suggestions that could help I nth e improvement of the manuscript:

We sincerely appreciate your positive and encouraging evaluation of our manuscript. Thank you for your kind support and constructive input throughout the review process

  1. Can there be a comparative study between different Gram-positive microbes to study aPDI efficacy? What was the reason to solely focus on mRSA?

    We appreciate your idea. There can definitely be a comparative study between different microbes as long as they exhibit similar binding sites for such photosensitizers to attach. We chose MRSA because it is the top pathogen responsible for severe infections, particularly in skin and soft tissues. Its growing resistance to multiple antibiotics makes it a clinically relevant and challenging target, highlighting the need for alternative strategies such as antibacterial photodynamic inactivation (aPDI). This explanation and clarification were included in the Introduction section.

  2. Also, a study was conducted with a single antibiotic. What was the rationale behind the single antibiotic test?

    We thank you for raising the question. There are three reasons for choosing vancomycin in this study: 1) Its chemical structure is amenable to modification without compromising the integrity of the molecule, as it contains a single primary amine that allows for selective conjugation; 2) It exhibits strong targeting ability against methicillin-resistant bacteria, such as MRSA, reducing the risk of losing enhanced targeting in our investigation; 3) In addition to enhancing aPDI effect, vancomycin may provide an antibiotic effect following light treatment, potentially preventing the recurrence of infections.

    A brief discussion has been added to the manuscript in Introduction section to address this matter:

    “Vancomycin was selected for this study for several reasons. Its chemical structure allows selective modification without compromising molecular integrity, due to the presence of a single primary amine that facilitates conjugation. It also exhibits strong targeting ability against methicillin-resistant bacteria such as MRSA, making it a reliable choice for our investigation. Furthermore, in addition to enhancing antibacterial photodynamic inactivation (aPDI), vancomycin may provide a delayed antibiotic effect after light treatment, offering the potential to prevent the recurrence of infections.”

  3. There are few limitations already mentioned by authors about TEM imaging. Apart from TEM imaging, what other methods were used to find the mechanism of Abx-dcPS penetration?

    We fully understand your point. Besides TEM, methods such as flow cytometry, fluorescence or confocal Z-stack imaging of biofilms, extracellular polymeric substance (EPS) disruption assays, atomic force microscopy, and mass spectrometry imaging are all promising approaches for tracking drug penetration. However, as a proof-of-concept technique, TEM was selected in this study because it provides a direct, straightforward, and cost-effective means of evaluation. By providing high-resolution images of the cell surface, TEM allows a more intuitive visualization of drug localization.

    A brief discussion has been added to the manuscript in Section 2.3.:

    “A combination of TEM with other techniques such as flow cytometry, fluorescence or confocal Z-stack imaging of biofilms, extracellular polymeric substance (EPS) disruption assays, atomic force microscopy, and mass spectrometry may allow us to observe intracellular localization of LC40e+. It is to be applied in the future investigation. In the present study, TEM has provided high-resolution images of the cell surface, allowing us to have a more intuitive visualization of drug distribution at the cell boundary.”

  4. Concentration of Abx-dcPs id too high 10-20μm. What is the range of optimising and balancing the safety and efficiency of the treatment?

    We fully understand your concern. An effective aPDI process relies on three key factors: the light source, photosensitizers, and molecular oxygen. In this study, we initiated the evaluation at a relatively lower light dosage combined with a higher photosensitizer concentration. The rationale for selecting this concentration was based on its toxicity profile in host HaCaT cells. At higher concentrations of Abx-dcPS (10–20 µM), some photocytotoxicity was observed in HaCaT cells, resulting in approximately a 1.0-log₁₀ reduction in viability, while MRSA cells were nearly eradicated. This concentration was therefore considered relatively safe for host cells. The concentration can be further reduced by increasing the light dosage or prolonging the irradiation time. We began our evaluation at the higher end of the concentration range to ensure the photosensitizers’ potential effectiveness. We fully acknowledge, however, that the optimal combination of aPDI agent concentration and light exposure warrants further investigation in future studies.

    A brief explanation has been added to the manuscript in Section 2.7.:

    “It is worth mentioning that an effective aPDI process relies on three key factors: the light source, photosensitizers, and molecular oxygen. In this study, we initiated the evaluation at a relatively lower light dosage combined with a higher photosensitizer concentration to ensure the photosensitizers’ potential effectiveness. The concentration can be further reduced by increasing the light dosage or prolonging the irradiation time. Therefore, optimal concentration of aPDI agent and light exposure should be investigated in future study.”

  5. aPDi activity on mRSA biofilms showed higher antimicrobial activity than planktonic cells. Is there any alternate strategy that has been tried to increase the efficiency of biofilm penetration?

    We appreciate the reviewer’s insightful comment regarding the relatively modest biofilm results. This observation is indeed consistent with the well-documented resistance of biofilms, largely attributed to the protective EPS matrix and limited penetration of therapeutic agents. In this study, although our primary aim was to establish proof-of-concept efficacy of LC40e+ against biofilm-associated MRSA, we still performed dose-dependent experiments, which higher inactivation of biofilms can be achieved at higher aPDI doses. However, we fully agree that additional strategies could further enhance activity. For example, several alternative strategies can be employed to improve penetration and efficacy against biofilms, including the use of biofilm-dispersing agents, combination therapy, and physical disruption techniques. Future work will include penetration studies using CLSM z-stack imaging to directly visualize distribution within biofilms, EPS disruption analysis to evaluate matrix destabilization, and combination approaches with dispersal agents to facilitate deeper penetration and improved bacterial inactivation.

    A note discussing these strategies has been added to the revised manuscript in Section 2.6.:

    “Although LC40e+ achieved significant activity against biofilm-associated MRSA (~2-log reduction), its efficacy seemed to be lower compared with planktonic cells (>6.5-log reduction). It should be noted that in a dose-dependent experiment, higher inactivation of biofilms can be achieved by applying higher aPDI doses. This outcome is consistent with the inherent resistance of biofilms due to their dense extracellular polymeric substance (EPS) matrix and restricted photosensitizer penetration. In future research, particular attention will be given to strategies to enhance biofilm eradication, including penetration analysis by CLSM z-stack imaging, EPS disruption assays to assess matrix destabilization, and combination approaches with biofilm-dispersal agents.”

  6. The non-linear graph showed a gradual decrease in ABMA concentration in the solution. Are there any precautions used by the authors?

    The singlet oxygen probe ABMA is a chromophore that exhibits sensitivity to light. The observed decrease in fluorescence intensity may arise from photodecomposition of ABMA by the light source itself, rather than exclusively from oxidative reactions caused by singlet oxygen. This issue is common among the most of similar commercial probes, and unfortunately, there is no straightforward method to completely eliminate it due to the direct interaction between light and the probe. Nevertheless, this photodecomposition was relatively minor compared to the decrease observed in the presence of photosensitizers, particularly within the first 100 min of irradiation. Given this, ABMA was primarily used as a blank control for comparison, and we did not pursue additional precautionary methods beyond this baseline.

  7. The material and method section like 4.8 lacks citations. Apart from a detailed description, it is advisable to add a citation for every methodology used.

    We thank the reviewer for the suggestion, The citation for the methodology we used has been added to the manuscript.

  8. In section 2.4 ABMA is previously synthesized. Is the synthesis method of the ABMA compound used for ROS is described in this study? There are various literature cited with 3 self-citations. Does it point to the synthesis of direct sources of compounds or previously synthesized ABMA was used?

    The singlet oxygen detecting probe ABMA was synthesized by our group, and its synthetic procedure can be directly linked to our previous publication [39]. This probe exhibits a relatively long shelf-life under dark conditions; therefore, previously prepared ABMA was used for the current evaluations.

    To facilitate reproducibility and allow readers to follow the procedure, a note was added to the manuscript in Section 4.1.:

    “The singlet oxygen detecting probe ABMA was prepared previously in our report [39].”

    Abrief synthetic protocol has been provided in the Supporting Information.

    “Preparation of tetrasodium α,α′-(anthracene-9,10-diyl)bis(methylmalonic acid salt) (ABMA)

    α,α′-(anthracene-9,10-diyl)bis(methylmalonic acid) (100 mg, 0.24 mmol) was suspended in H2O (5.0 mL), with sodium hydroxide (0.05 g), followed by ultrasonication for a period of 5.0 min to give a clear solution. The tetrasodium salt product was precipitated upon the addition of ethanol. The solids were washed three times with ethanol and dried in vacuo to yield ABMA in 81% (97 mg). Spectroscopic data: FT-IR (KBr) νmax 3446 (vs, water peak), 2951 (w), 2923 (w), 2892 (w), 2844 (w), 1592 (vs), 1423 (m), 1339 (m), 1317 (s), 892 (m), 815 (w), 757 (m), 695 (w), 628 (w), 594 (w), 517 (w) cm−1. UV–vis (H2O, 1.0 × 10−5 M) λmax 327 (shoulder band, ε = 9.62 × 105), 344 (ε = 2.54 × 106), 361 (ε = 5.65 × 106), 380 (ε = 9.38 × 106) and 402 (ε = 9.12 × 106 cm2/mol) nm; 1H NMR (500 MHz, D2O, ppm) δ 8.46 (d, 4H), 7.62 (dd, 4H), 4.16 (d, 4H), and 3.56 (t, 2H); 13C NMR (500 MHz, D2O, ppm) δ 179.00 (4C), 132.37 (2C), 129.40 (4C), 125.53 (2C), 125.41 (2C), 125.36 (2C), 125.30 (2C), 59.77 (2C), and 28.29 (2C).”

  9. What are the main challenges raised in this study, and why did the VCMe-mChlPd-N10+ (LC40e+) conjugate combine vancomycin with a decacation strategy to target and inactivate methicillin-resistant Staphylococcus aureus (MRSA) selectively? Furthermore, what important advantages does this combination approach offer compared to conventional antibiotics or isolated antimicrobial photodynamic inactivation (aPDI) treatments?

    The development of LC40e+ is based on an evolutionary series of mesochlorin conjugates. Our earlier research demonstrated that varying the number of positive charges (from 2 to 15) on the mesochlorin scaffold significantly impacts antibacterial activity, with 10 charges yielding the highest efficacy. In order to improve its targeting ability further, a new targeting mechanism is necessary to be introduced, we subsequently introduced vancomycin (VCM) as a synergistic targeting component for enhancing aPDI. However, when directly conjugate VCM to mesochlorins, due to the large steric hindrance of VCM, its accessibility to the MRSA cell surface was greatly reduced, resulting in minimal aPDT efficacy. Building on these findings, in the present manuscript we designed and studied a combinatorial aPDT–antibiotic compound, LC40e+, a molecule incorporating 10 charges, mesochlorin, a short polymer linker, and vancomycin, along with new biological data demonstrating its selective binding and potent antimicrobial photodynamic effects against MRSA.

    The molecule was specifically designed to selectively target MRSA, the role of cationic charges is to provide targeting ability through electrostatic interactions with negatively charged components of bacterial cell walls, while vancomycin specifically binds to D-Ala–D-Ala residues in the peptidoglycan precursors of Gram-positive bacteria, offering a distinct and complementary mode of interaction compared with cationic charges. The combination of the two offers enhenced selective binding on MRSA.

    A discussion has been added to the manuscript in Discussion section to address this mater:

    “Furthermore, in our early studies, we focused exclusively on the role of cationic charges, which provide targeting ability through electrostatic interactions with negatively charged components of bacterial cell walls. Specifically, lipopolysaccharides bearing phosphate and carboxylic acid anions dominate the outer membrane of Gram-negative bacteria, while teichoic acids and terminal carboxylic acids contribute to the strong negative charge of the peptidoglycan layer in Gram-positive bacteria. These interactions guided the initial design of our mesochlorin conjugates. At a later stage, to move beyond purely static electrostatic interactions, we introduced antibiotics as synergistic targeting components to enhance aPDI through an alternative binding mechanism…………….offering a distinct and complementary mode of interaction compared with cationic charges [40, 56]. While the mechanistic contributions of vancomycin and cationic charges could, in principle, be examined individually, such separate evaluations would not provide direct evidence of the synergistic effect resulting from their combination. As is often observed in synergistic systems, the combined effect can exceed the sum of the individual contributions. Therefore, we strategically advanced to evaluating conjugates that integrate both cationic charges and antibiotic moieties, allowing us to directly assess their combined targeting potential in enhancing aPDI efficacy.”

    The synergistic strategy we developed provides several advantages over conventional antibiotics or aPDT alone. Unlike traditional antibiotics, which require time to exert their effects and lead to gradual cell rupture, our approach achieves rapid and effective bacterial killing in a single step, reducing the opportunity for resistance development. In addition to enhancing aPDI effect, this strategy may offer a delayed antibiotic effect following light treatment, potentially preventing the recurrence of infections. Moreover, A molecular combination of aPDI agent and antibiotic should also be able to provide covalent co-delivery of two drug components concurrently, leading to potential significant improvement in bioactive functions, Finally, the inherent photophysical properties of the chlorin scaffold, particularly its strong red fluorescence emission, provide the potential for real-time visualization and tracking of photosensitizer delivery.

    The discussion was presented in the Conclusion section of the manuscript.

Round 2

Reviewer 3 Report

Comments and Suggestions for Authors

Accept in present form.